# A Review on Ionic Liquid Gas Separation Membranes

**DOI:** 10.3390/membranes11020097

**Published:** 2021-01-30

**Authors:** Karel Friess, Pavel Izák, Magda Kárászová, Mariia Pasichnyk, Marek Lanč, Daria Nikolaeva, Patricia Luis, Johannes Carolus Jansen

**Affiliations:** 1Department of Physical Chemistry, University of Chemistry and Technology Prague, Technická 5, 166 28 Prague, Czech Republic; karel.friess@vscht.cz (K.F.); izak@icpf.cas.cz (P.I.); marek.lanc@vscht.cz (M.L.); 2Czech Academy of Sciences, Institute of Chemical Process Fundamentals, Rozvojová 135, 165 02 Prague, Czech Republic; karaszova@icpf.cas.cz (M.K.); pasichnyk@icpf.cas.cz (M.P.); 3Materials & Process Engineering, UCLouvain, Place Sainte Barbe 2, 1348 Louvain-la-Neuve, Belgium; daria.nikolaeva@uclouvain.be (D.N.); patricia.luis@uclouvain.be (P.L.); 4Institute on Membrane Technology (CNR-ITM), Via P. Bucci, 17/C, 87036 Rende (CS), Italy

**Keywords:** gas separation, ionic liquid, polymerized ionic liquids, ionic liquid blends, ion gel membrane, transport properties

## Abstract

Ionic liquids have attracted the attention of the industry and research community as versatile solvents with unique properties, such as ionic conductivity, low volatility, high solubility of gases and vapors, thermal stability, and the possibility to combine anions and cations to yield an almost endless list of different structures. These features open perspectives for numerous applications, such as the reaction medium for chemical synthesis, electrolytes for batteries, solvent for gas sorption processes, and also membranes for gas separation. In the search for better-performing membrane materials and membranes for gas and vapor separation, ionic liquids have been investigated extensively in the last decade and a half. This review gives a complete overview of the main developments in the field of ionic liquid membranes since their first introduction. It covers all different materials, membrane types, their preparation, pure and mixed gas transport properties, and examples of potential gas separation applications. Special systems will also be discussed, including facilitated transport membranes and mixed matrix membranes. The main strengths and weaknesses of the different membrane types will be discussed, subdividing them into supported ionic liquid membranes (SILMs), poly(ionic liquids) or polymerized ionic liquids (PILs), polymer/ionic liquid blends (physically or chemically cross-linked ‘ion-gels’), and PIL/IL blends. Since membrane processes are advancing as an energy-efficient alternative to traditional separation processes, having shown promising results for complex new separation challenges like carbon capture as well, they may be the key to developing a more sustainable future society. In this light, this review presents the state-of-the-art of ionic liquid membranes, to analyze their potential in the gas separation processes of the future.

## 1. Introduction

The term ‘Ionic Liquids’ (IL), in the most modern sense, describes the molten salts whose physical state can be described as a liquid at temperatures below 100 °C [1,2]. While first discovered for military battery applications, ILs swiftly found their way into other research fields such as solvent media for organic reactions, catalysts, lubricants, and separation media [3,4,5]. Most importantly, ILs entirely consist of cations and anions in the absence of any molecular solvents. Both anions and cations can be represented by organic and inorganic ions interchangeably. The rapid growth in research and academic application of ILs facilitated the diversification of ions used [5]. Although the synthetic flexibility of ILs allows almost inexhaustible combinations of ions comprising them, the most widely used cations and anions can be easily identified, based on published reports (Figure 1).

At the present stage of membrane science development, the application of ionic liquid membranes (ILMs) in technologies of gas separation has received great attention due to their low vapor pressure, high thermal resistance, and high electrochemical stability. Generally, ILMs are formed by combining large organic cations with a wide range of anions. It is essential to immobilize the ILs to support materials in separation processes, which minimizes the amount of active phase required for this process and greatly facilitates the recovery and reuse of ILs. These membranes can be easily used on an industrial scale to separate hydrogen from ammonia purge gas [6], to purify natural gas [7], to obtain oxygen or inert gases, to separate volatile organic compounds (VOCs) from gas streams [8], to capture CO_2_ from flue gases [9], or for recovery and enrichment of biohydrogen [10]. Besides, the disadvantages of IL, such as high viscosity, high production cost, unclear toxicity, and potential environmental impact, can be modified via alteration of cations and anions. Therefore, studies aimed at improving and developing new types of IL-based gas separation membranes are quite relevant. Tailor-made ILMs can be synthesized using different matrices to support the IL, forming a robust membrane that acts as a separator between the gas supply and the receiving phase. 

In recent years, it was shown that immobilizing and modifying the solid supports with ILs are efficient ways to exploit IL capability in the adsorptive removal processes [11]. As a result, there are various combinations which make it possible to significantly improve the basic characteristics of membranes and use them in energy-saving and environmentally friendly technologies. The present review highlights the current state of the art in ionic liquid-based membranes for gas separation purposes. Among the main types of ionic liquid membranes, one can distinguish three main functional concepts (Figure 2) that will be discussed more closely within the body of the review:

RTILs (supported IL membranes, SILMs), which offer the most straightforward approach for ILs separation performance testing at the expense of long-term process stability;

PILs, which resolve the stability issues of SILMs at the expense of curbed flux;

PIL/IL blends and conventional polymer/IL blends, which serve as a compromise between the two abovementioned IL-based materials.

The present manuscript aims to provide a comprehensive overview of the current state of the art in ionic liquid-based membranes for gas separation applications, to highlight the pros and cons of each of these three different categories of ILMs, and to compare their performance.

## 2. Supported Ionic Liquid Membranes

Supported liquid membranes, along with other types of liquid membranes, existed before the ionic liquids appeared [12,13,14,15], but the boom of ionic liquids with their unique physical and chemical properties made supported liquid membranes interesting from the practical point of view. To overcome the disadvantages of supported liquid membranes, scientists started to use ionic liquids instead of conventional solvents. Supported ionic liquid membranes (SILMs) have been the object of interest of many research groups for the last twenty years, as highlighted by the number of published reviews [16,17,18,19,20,21,22,23,24]. The new types of SILMs created have high selectivity and permeability, and they combine the advantage of the stripping process by liquids with high solubility and the easy operation of membranes. 

### 2.1. Structures and Membrane Preparation Techniques

#### 2.1.1. Precursor Materials and Properties

Typical supports are hydrophilic, but there is limited understanding of the effect of different membrane supports with diverse chemical nature on gas permeability and selectivity for gas mixtures. Long-life SILMs membranes with high stability can be produced by using hydrophobic material, which increases the thickness of the boundary layers on the interfaces of SILMs mainly due to the physical bonding between ILs and substrates when ILs are coated in the pores of the substrates [25]. SILM supports are mainly differentiated based on their geometry (hollow fiber or flat sheet), pore size and homogeneity (symmetric/asymmetric) [26,27,28,29], and surface free energy (hydrophobicity/hydrophilicity) [30]. All these parameters have a direct effect on the successful IL deposition within the support pores. The support compatibility is also largely affected by ILs, which can significantly influence the preparation process through their nature, viscosity [27,31], surface tension, physical state at a given temperature, and stability in the atmosphere.

The integrity of SILMs relies on the physico-chemical properties of the materials used, which can be both organic and inorganic. While inorganic supports are mostly ceramic or graphene oxide-based [32,33,34], other materials include molybdenum disulfide and activated carbon [35]. Inorganic supports offer superior temperature stability, and since ILs are known for their high degradation temperatures, this type of SILM may be used in processes operating at elevated temperatures. Organic supports can be freely chosen to ensure good compatibility between the IL and pore walls of the support. The majority of reported materials include commercially available or in-house produced Polyvinylidene fluoride (PVDF) [28,30,36,37,38,39], PAN [40], tetrafluoroethylene-vinylidene fluoride co-polymer [41], and polyimide [42,43]. Polymeric supports have a lower retail price, providing a higher incentive for their application in low-temperature processes. In addition, the low cost of polymeric supports allows the extensive study of their physico-chemical and geometrical properties via destructive techniques, such as SEM, AFM, TGA, tensile tests, etc. Composite supports comprising inorganic alumina coating with a thin mesoporous polymer layer and carbon-based nanocomposites have been suggested to improve the IL stability within the pores of the support [44,45,46].

#### 2.1.2. Membrane Preparation

SILMs can be prepared using chemical or physical methods [47] including immobilization of an ionic liquid by impregnation [48], sol-gel method [49], encapsulation [50], and immobilization via covalent bonding [51]. For the two major preparation techniques, coating [33,40] and soaking, the prevailing processes rely on pressure or vacuum-assisted infiltration [32,34,39,41]. Infiltration techniques are trapping ILs inside the pores, based on their surface tension and viscosity, and require the estimation of the capillary forces. Usually, SILMs prepared by the pressure method have the highest amounts of immobilized ionic liquid [52]. The property changes of the membranes prepared under pressure or vacuum are quite dependent on the viscosity of the embedded ionic liquids [16]. The porous structure of the supported membrane leads to the progressive loss of the liquid embedded in the membrane into the surroundings, and these losses of IL were found to be higher when immobilization was carried out under vacuum. The vacuum method resulted in a lower amount of ILs being impregnated when the viscosity of ILs increased. Immobilization under vacuum has the advantage that it is a relatively easy preparation method, but is most suitable for ILs of low viscosity [52]. Hernández-Fernández and coworkers found that the preparation method of SILM has a connection with membrane stability. For example, SILMs prepared by the vacuum method show higher losses of the IL during operation, and the relatively high losses increase with increasing viscosity of the IL. SILMs with the highest-stability can be achieved by using the pressure method, in which case the pores of the support material can be completely filled with the ILs. Moreover, it was found that the methods of direct immersion and vacuum are suitable for the low-viscosity ILs, while the pressure method is preferred for high-viscosity ILs [20]. Supported ionic liquid membranes (SILMs) were also prepared using a supercritical fluid deposition (SCFD) method at 12 MPa and 50 °C. The IL easily diffused into the small pores, helped by the capillary condensation in the presence of ethanol besides supercritical CO_2_, leading to the selective filling of the meso- and micropores in the supports [53]. Figure 3 shows the most common configurations of SILMs: A porous support, impregnated with IL, or a quasi-solidified IL phase [20]. Both configurations can exist with or without additional silicone coating.

#### 2.1.3. Long Term Stability

The susceptibility of SILMs to leakage of the ionic liquid from the larger pores of the macroporous support results in process instability in the long run [13]. Recent pilot-scale trials at three power plants in Germany revealed that optimized SILM could maintain their structural and process stability for more than 335 h in the absence of SO_3_-containing aerosols [40]. However, the presence of aerosol was detrimental to the operation stability and separation performance of SILMs. Material optimization in this area is primarily directed towards the improved compatibility between the IL and support used. This optimization can be achieved through surface treatment (e.g., with plasma) [42], implementation of mesoporous supports with uniform pore sizes [34,45], and by a specific membrane preparation process [54]. The successful compatibility enhancement, as determined by Scanning Electron Microscopy with Energy Dispersive X-ray Analysis SEM-EDX and X-ray Photoelectron Spectroscopy (XPS) [52,55] analysis, will usually yield prolonged operation times without performance deterioration [37,56] or SILM weight loss [29,57,58]. Alternatively, the bubble-point or breakthrough pressure measurement [42,54] allows a direct measurement of the critical gas pressure required to displace a gas from the porous support containing the IL. This method allows for fast product and process optimization steps during the SILM development without the necessity to perform elaborate pilot testing in a gas separation setup.

### 2.2. Pure Gas Transport Properties

A supported ionic liquid membrane is the simplest configuration for testing the transport properties of a particular ionic liquid for gases. The liquid is trapped inside the pores of porous support by capillary forces. Of course, the support has to be well wetted by the ionic liquid and should be chemically inert. The number of tested support materials is quite high [59] and includes, besides porous polymer membranes, 3D graphene with excellent CO_2_/CH_4_ separation performance [60]. The solution diffusion model of transport is usually applied. The diffusivity of a gas through the ionic liquid is often correlated with the viscosity of the ionic liquid [61,62,63,64,65]. The dissolution of gases in ionic liquids follows the regular solution theory [66,67,68,69,70]. The selectivity may be correlated with the molar volume of the ionic liquid [71,72]. Some works deal with quantum chemistry of interactions between the ionic liquid and the permeating gas to provide a prediction of the permeability of the SILM. COSMO-based models seem to be the most accurate and widely applicable [73,74,75,76], and they were successfully used to predict gas permeability and selectivity of ionic liquid membranes. An innovative approach to the physico-chemical basics of interactions between ionic liquids and permeating gases [77,78,79] used a correlation between the transport properties of the ionic liquid (permeance, selectivity) and not only viscosity, but also the quantum chemistry value described as the “hydrogen bond acceptance ability”. The results, especially in terms of permeance, are still not fully convincing and need further research, but if such an approach is verified, it will help to choose the right ionic liquid for a particular application. It must be noted that the transport properties of SILMs are often expressed as permeance (unit GPU). Whereas, for homogeneous membranes the permeability is usually defined as the permeability coefficient, a material-specific property, normalized for the thickness (with unit Barrer, where 1 Barrer = 10^−10^ cm^3^_STP_ cm cm^−2^ s^−1^ cmHg^−1^ = 3.348 × 10^−16^ mol m m^−2^ s^−1^ Pa^−1^), this is not possible for non-homogeneous or composite materials with undefined or unknown thickness, such as SILMs or thin-film composites. The permeance is usually expressed in GPU ‘Gas Permeance Unit’, where 1 GPU = 1 Barrer µm^−1^ = 10^−6^ cm^3^_STP_ cm^−2^ s^−1^ cmHg^−1^ = 3.348 × 10^−10^ mol m^−2^ s^−1^ Pa^−1^. Alternatively, it is often expressed in m^3^_STP_ m^−2^ h^−1^ bar^−1^. Permeance is a membrane property and not a material property

The transport mechanism of a gas through SILMs can be described by the solution–diffusion model (S–D model): The gas dissolves in the IL at the feed side, diffuses through the IL layer due to a pressure difference across the membrane, desorbs from the permeate side, and is swept away by the permeate stream at low pressure. The concept of facilitated transport for CO_2_ involves the addition of a mobile chemical to the SILM, a so-called carrier, that can reversibly bind to CO_2_. This would enhance the solubility of only CO_2_, relative to other gases, increasing the membrane selectivity for CO_2_ [62]. Some recent data obtained for ionic liquids in gas separations are summarized in Table 1, Table 2, Table 3 and Table 4 at the end of this section. Looking closer at the performed experiments and obtained data, one can see:CO_2_/N_2_ and CO_2_/CH_4_ mixtures are the most popular for the experiments (Table 1 and Table 2);emerging alternatives are for instance alkene/alkane separation;the ionic liquids and supports are not changing significantly during the time;porous PVDF and Polyethersulfone (PES) are leading supports, although alternatives are possible [80,81]; andsometimes there are some trials for some more exotic anions like those based on amino acids or cholinium cations, but imidazolium-based ILs with the Tf_2_N anion dominate.

Some research groups tried to enhance the performance of supported ionic liquid membranes by adding some particles or by mixing different ionic liquids. Goueveia et al. [82] added ionic liquids with amino acid anions (glycinate, prolinate, etc.) to [EMIM][C(CN)_3_]. The ionic liquid mixtures had higher CO_2_ solubility, but the permeability of CO_2_ decreased compared to that in pure [EMIM][C(CN)_3_]. Another attempt to enhance the gas transport properties of [EMIM][Tf_2_N] and [BMIM][DCA] ionic liquids via the addition of polyamidoamine dendrimer was presented by Jie et al. [83]. 

SILMs represent a good set-up for basic research dealing with the solubility of gases in ionic liquids and for studying basic physico-chemical interactions between gases and ionic liquids, but most membranes are tested only with pure gases or model binary (max. quaternary [84]) mixtures, while application of real gases is rather exceptional [85]. Up to now, there have been no reliable ways to fabricate a stable supported ionic liquid membrane with mechanical properties convenient for use in real gas separation conditions under high pressure with multi-component mixtures, and thus researchers should be careful when declaring the high potential of these membranes for application in CO_2_/N_2_ separation. The ionic liquids and used supports are still expensive materials and are not produced in large enough amounts. One issue consists of the scale-up of membrane preparation, which is in progress [86], but has not been fully solved yet. Another repeating problem is the stability of SILMs [43]. To the best of our knowledge, the only pilot-scale tests with real supported ionic liquid membranes were done by Klingberg et al. [40], who tested such membranes in conditions of powerplant as a method of flue gas separation, and identified problems with leakage of the ionic liquid from the membrane. For large-scale gas separation applications, the more stable configurations of ionic liquid membranes described in other parts of this work are more convenient.

Separation of CO_2_ from H_2_, N_2_, or CH_4_ by SILMs has economic benefits, as SILMs require lower energy consumption. New membranes based on graphene oxide, confining the [BMIM][BF_4_] ionic liquid, show a good improvement of the selectivity for CO_2_ over other gases [32]. Only few researches are dedicated to the selective separation of H_2_S and CH_4_ or H_2_S and CO_2_ by SILMs. Zhang et al. first reported about facilitated transport of H_2_S in SILMs based on [BMIM][Ac], which helped with the selective separation of H_2_S from natural gas, as they demonstrate the highest H_2_S/CO_2_ selectivity [87]. Cichowska-Kopczynska presented highly CO_2_ selective membranes for CO_2_/CH_4_ separation, based on the 1-alkyl-3-methylimidazolium cation, using polypropylene as the support [88]. These were found to be stable and effective in the lower temperatures range, 283–298 K.

#### 2.2.1. Effect Ionic Nature

Tomé et al. [89] concluded that the type of anion has a stronger influence on the gas solubility than the type of cation. Cholinium-based SILMs with carboxylate ionic liquids show high permselectivity for CO_2_/CH_4_ and CO_2_/N_2_. Polydimethylsiloxane (PDMS) membranes with 1-ethyl-3-methylimidazolium acetate ([EMIM][Ac]) ionic liquid shows high CO_2_/N_2_ selectivities up to 152 [44]. Recently, Shamair et al. created SILMs based on Benzimidazolium-1-acetate as ionic liquid and polyimide support [9]. It was found that the nucleophilic acetate ion with high complexation energy favors the separation of CO_2_, giving high CO_2_/CH_4_ and CO_2_/N_2_ selectivities of 37.92 and 40.29, respectively. SILM based on protic ionic liquid (3-aminopropyl) trimethoxysilane and acetic acid showed high selectivity of 41 for CO_2_ removal from mixed gas CO_2_/CH_4_. The main drawback of such membranes is that for good selectivity they need quite high temperatures. On the other hand, they can display high mechanical stability [90]. It was confirmed that a polyvinylidene fluoride porous membrane with the acetate-based RTILs showed a negligible effect of the cation on the CO_2_/N_2_ gas permeability [91]. The CO_2_ permeance values for these membranes are in the order of 852–2114 Barrer, and CO_2_/N_2_ selectivity ranges between 26 and 39.

#### 2.2.2. Effect of Active Groups 

It was shown that branched alkyl side chains and trimethylsilyl end-groups prevent crystallization of the IL in the SILMs and lead to higher viscosities and lower densities [31], suitable for CO_2_/N_2_ separation. Zhang, X., et al. [92] designed imidazolium-based phenolate ionic SILMs that have dual-site interaction centers to isolate CO_2_ from N_2_ due to the facilitated transport mechanism (transfer of CO_2_ from carbene to phenolate anion). In a comparison of different ILs [93], the presence of a siloxane group in the cation was shown to create SILMs with high CO_2_/N_2_ permselectivity without changing the CO_2_/CH_4_ selectivity. SLIMs based on 1-alkyl-3-methylimidazolium acetate showed high final purity (93.3–95.98%) in CO_2_/CH_4_ mixed gas separation, and the selectivity increased with the number of carbon atoms in the 1-alkyl side chain. Although this type of ionic liquid is stable, free of halogen, and has a low price, a disadvantage is a slightly lower CO_2_/CH_4_ selectivity [94].

#### 2.2.3. Effect of Viscosity

Green and cost-effective SILMs based on [Choline][Pro]/polyethylene glycol 200 (PEG200) mixtures for CO_2_/N_2_ separation were developed and showed that the CO_2_ permeability increased three times by decreasing the viscosity of the ionic liquids from 370 to 38 mPa·s [95].

#### 2.2.4. Effect of the Water Vapor Content in the Gas Stream

The presence of water influences the stability of SILMs due to the formation of water clusters within the ionic liquid. Porous anodic alumina membranes with pore size 20–100 nm were mixed with different anions. SILMs showed a high CO_2_ permeability of more than 1000 Barrer and an ideal CO_2_/N_2_ selectivity between 12 and 21. It was reported that the addition of water (10%) to the SILMs increased the CO_2_ and N_2_ permeance due to the decrease in ionic liquid viscosity. A higher water content decreases the CO_2_ solubility due to hydrogen bonding between water and the ionic liquid, limiting the interactions between CO_2_ and the anion [96].

#### 2.2.5. Effect of Temperature

Four supported ionic liquid membranes (SILMs) were prepared by fixing ionic liquids (ILs) 1-butyl-3-methylimidazolium dicyanamide ([BMIM][DCA]), acetate ([BMIM][AC]), trifluoromethanesulfonate ([BMIM][TfO]), and bis((trifluoromethyl)sulfonyl)imide ([BMIM][Tf_2_N]), respectively, in polyvinylidene fluoride membranes, and were used for the separation of CO_2_/H_2_ and CO_2_/N_2_ gas mixtures. The gas permeation rate of the SILMs increased with the increasing temperature from 30 to 50 °C. This is mainly ascribed to the accelerated thermal motion of the gas molecules with increasing temperature because the solubility of the gas molecules in the IL decreases. The permeation of CO_2_, H_2,_ and N_2_ in the SILMs fit the solution-diffusion mechanism [97]. It has been found that the selectivity of CO_2_/N_2_ and CO_2_/CH_4_ systems on the membranes based on [BMIM][BF_4_] slightly decreases with temperature [98].

#### 2.2.6. Effect of Pressure

In the SILMs with diamine-monocarboxylate-based protic ionic liquids were made for selective separation of the CO_2_/N_2_ and CO_2_/CH_4_ gas pairs under humidified conditions. The permeability in the system increases with decreasing transmembrane pressure [99].

### 2.3. Mixed Gas Transport Properties

Althuluth et al. [100] reported that the pure gas permeability of various natural gas components (CO_2_, CH_4_, C_2_H_6_, and C_3_H_8_) through SILMs based on an inorganic support at a trans-membrane pressure of 0.7 MPa and temperature of 313 K follows the order *P*_CO2_ > *P*_CH4_ > *P*_C2H6_ > *P*_C3H8_. CO_2_ had a higher permeability than the rest of the gases due to better solubility in the RTIL. The mixed gas selectivity of CO_2_/CH_4_ (50/50%, *v*/*v*) was found to be much lower (*α* = 1.15) than the ideal permselectivity (*α* = 3.12). Remarkably, CH_4_ showed a much higher diffusivity than CO_2_, which explains why the permselectivity of a binary CO_2_/CH_4_ mixture is so much lower than the solubility selectivity of pure CO_2_ and CH_4_. Scovazzo et al. carried out an extensive study on mixed gas selectivity, non-humid operation, and long-term operation of SILM for the CO_2_/CH_4_ gas pair [101]. They concluded that the mixed gas selectivity did not decrease in dry conditions for more than 106 days. In another study, Hojniak et al. studied a series of RTILs with triethylene glycol appendages attached to cations, and tested them for CO_2_/CH_4_ and CO_2_/N_2_ gas pairs. They reported a 2.3 times higher mixed gas selectivity than RTIL without nitrile group [102]. Neves et al. investigated the effect of hydrophobic support and different cations in SILMs [57], showing that increasing the alkyl chain length of the cation resulted in a small decrease in the selectivity of mixed gases compared to pure gases.

**Table 1 membranes-11-00097-t001:** Some recent data for supported ionic liquid membranes in CO_2_/N_2_ separations.

Support	Ionic Liquid	*P*CO_2_ [Barrer] ^(a)^	*α* (CO_2_/N_2_) [-]	Ref.
Nanoporous alumina	[EMIM][Tf_2_N]	44 GPU	4	[96]
	[EMIM][TfA]	43 GPU	4.5	
	[BMIM][Tf_2_N]	44 GPU	4.4	
	[C_3_C_1_MIM][Tf_2_N]	30 GPU	3.0	
	[BMIM][Ac]	54 GPU	5.4	
Porous Polyvinylidene fluoride (PVDF)	[EMIM][Ac]	878.8	33.7	[91]
	[BMIM][Ac]	851.9	34.6	
	[Vbtma][Ac]	1100	39	
Polydimethylsiloxane (PDMS)/ceramic composite	[EMIM][Ac]	2000	152	[44]
Polymer support	[Si-C_1_-C_3_-MIM][Tf_2_N]	311	25	[31]
Polyethersulfone (PES)	[BMIM][Pho]	2540	127	[92]
Porous alumina + phenolic resin	[EMIM][BF_4_]	0.182 GPU	40	[45]
Hydrophilic PES	[Choline][Pro]/PEG200 1:0	343	35	[95]
	[Choline][Pro]/PEG200 1:1	810	28	
	[Choline][Pro]/PEG200 1:2	1138	12	
Hydrophilic Polytetrafluoroethylene (PTFE)	[Choline][Lev]	18	42	[89]
	[Choline][Lac]	7	45	
	[Choline][Gly]	6	50	
	[Choline][Mal]	2	40	
Graphene oxide	[BMIM][BF_4_]	68.5 GPU	382	[32]
MFFK—1 membrane	[BMIM][BF_4_]	103	10	[98]
	[BMIM][doc]	57	5	
Hydrophilic PES	[DMAPAH][EtOAc]	3028	151	[99]
	[DMAPAH][TFA]	3352	168	
PVDF	[(SiOSi)C_1_MIM][Tf_2_N]	545	24	[93]
	[(SiOSi)C_1_MIM][C(CN)_3_]	568	37	
Hydrophilic PVDF	[BMIM][DCA]	4788	6	[97]
	[BMIM][Ac]	520	23	
	[BMIM][TfO]	857	14	
	[BMIM][Tf_2_N]	3883	6	

^(a)^ Permeability in Barrer, unless stated otherwise (1 Barrer = 10^−10^ cm^3^_STP_ cm cm^−2^ s^−1^ cmHg^−1^ = 3.348 × 10^−16^ mol m m^−2^ s^−1^ Pa^−1^); permeance in GPU (Gas Permeance Unit) (1 GPU = 1 Barrer µm^−1^ = 10^−6^ cm^3^_STP_ cm^−2^ s^−1^ cmHg^−1^ = 3.348 × 10^−10^ mol m^−2^ s^−1^ Pa^−1^). The permeability (better: Permeability coefficient) is normalized for the membrane thickness and is a coefficient, defining the material properties. Instead, the permeance is not normalized for the membrane thickness and is often used for thin-film composite membranes, or for supported liquid membranes, in which the precise thickness is often unknown.

**Table 2 membranes-11-00097-t002:** Some recent data for supported ionic liquid membranes in CO_2_/CH_4_ gas separations.

Support	Ionic Liquid	*P*i [Barrer] ^(a)^	*α* (CO_2_/CH_4_) [-]	Ref.
Polypropylene	[EMIM][Tf_2_N]	636	40.4	[88]
	[EMIM][TfO]	1076	32.6	
	[BMIM][Tf_2_N]	936	74.3	
	[BMIM][TfO]	754	48.1	
Hydrophilic PTFE	[Ch][Lev]	18	20	[89]
	[Ch][Lac]	7	21	
	[Ch][Gly]	6	35	
	[Ch][Mal]	2	25	
Graphene oxide	[BMIM][BF_4_]	68.5 GPU	234	[32]
PVDF	[BMIM][BF_4_]	47 × 10^−11^ m^2^ s^−1 (b)^	24	[94]
PES	[BMIM][BF_4_]	42	23	
PVDF	[BMIM][Ac]	38	18	[94]
PES	[BMIM][Ac]	33	15	
PVDF	[OMIM][Ac]	66	21	[94]
PES	[OMIM][Ac]	60	19	
P-84 polyimide	[APTMS][Ac]	23 GPU	41	[90]
MFFK—1 membrane	[BMIM][BF_4_]	103	10	[98]
	[BMIM][doc]	57	5	
hydrophilic PES	[DMAPAH][EtOAc]	3028	72	[99]
	[DMAPAH][TFA]	3352	67	
PVDF	[(SiOSi)C_1_MIM][Tf_2_N]	545	10	[93]
	[(SiOSi)C_1_MIM][C(CN)_3_]	568	11	
Hydrophilic PVDF	[BMIM][DCA]	4788	2.5	[97]
	[BMIM][Ac]	520	5	
	[BMIM][TfO]	857	11	
	[BMIM][Tf_2_N]	3883	14	
γ-Alumina	[EMIM][FAP]	208 GPU	3	[100]

^(a)^ Permeability of the more permeable gas in Barrer unless stated otherwise (1 Barrer = 10^−10^ cm^3^_STP_ cm cm^−2^ s^−1^ cmHg^−1^ = 3.348 × 10^−16^ mol m m^−2^ s^−1^ Pa^−1^); permeance in GPU (1 GPU = 1 Barrer µm^−1^ = 10^−6^ cm^3^_STP_ cm^−2^ s^−1^ cmHg^−1^ = 3.348 × 10^−10^ mol m^−2^ s^−1^ Pa^−1^). ^(^^b)^ Authors call it permeability, but the units correspond to the diffusivity.

**Table 3 membranes-11-00097-t003:** Recent data for supported ionic liquid membranes in CO_2_/H_2_ and H_2_S/CO_2_ separations.

Mixture, i/j	Support	Ionic Liquid	*P*i [Barrer] ^(a)^	*α* (i/j) [-]	Ref.
CO_2_/H_2_	Graphene oxide	[BMIM][BF_4_]	68.5 GPU	24	[32]
H_2_S/CO_2_	PVDF	[BMIM][Ac]	5279	11.9	[87]
		[BMIM][BF_4_]	3708	3.5	
H_2_S/CH_4_	PVDF	[BMIM][Ac]	5279	142	[87]
		[BMIM][BF_4_]	3708	140	

^(a)^ Permeability of the more permeable gas in Barrer unless stated otherwise (1 Barrer = 10^−10^ cm^3^_STP_ cm cm^−2^ s^−1^ cmHg^−1^ = 3.348 × 10^−16^ mol m m^−2^ s^−1^ Pa^−1^); permeance in GPU (1 GPU = 1 Barrer µm^−1^ = 10^−6^ cm^3^_STP_ cm^−2^ s^−1^ cmHg^−1^ = 3.348 × 10^−10^ mol m^−2^ s^−1^ Pa^−1^).

**Table 4 membranes-11-00097-t004:** Some recent data for supported ionic liquid membranes in volatile organic compound (VOC) separations and olefin/paraffin separations.

Mixture, i/j	Support	Ionic Liquid	*P*i [Barrer] ^(a)^	*α* (i/j) [-]	Ref.
H_2_O/Air	Alumina on PVDF	[EMIM][BF_4_]	26,000	1000	[103]
		[EMIM][DCA]	46,000		
		[EMIM][ESU]	46,000		
Ethylene/ethane	PVDF	CuCl/[BMIM][Cl]	3500	9	[104]
Ethylene/ethane	PVDF	[EMIM][Me_2_PO_4_] +Ag	100	35	[105]
		[EMIM][Et_2_PO_4_] + Ag	80	40	
Propylene/propane	Porous alumina	[HMMIM][Tf_2_N] + Ag carriers	280	9	[106]
		[HMIM][Tf_2_N] + Ag carriers	450	7	

^(a)^ Permeability of the more permeable gas in Barrer unless stated as permeance in GPU. (1 Barrer = 10^−10^ cm^3^_STP_ cm cm^−2^ s^−1^ cmHg^−1^ = 3.348 × 10^−16^ mol m m^−2^ s^−1^ Pa^−1^; 1 GPU = 1 Barrer µm^−1^ = 10^−6^ cm^3^_STP_ cm^−2^ s^−1^ cmHg^−1^ = 3.348 × 10^−10^ mol m^−2^ s^−1^ Pa^−1^).

## 3. Polymeric Ionic Liquids (PILs)

Polymeric ionic liquids (PILs) represent a new kind of polyelectrolyte polymers with a cation or anion on the repeating units. PILs are usually synthesized from IL monomers and thus contain the same moieties as the ILs. PILs preserve many of the beneficial properties of ILs such as thermal and chemical stability and ion conductivity. Moreover, PILs benefit from the intrinsic polymer properties [23,107,108]. In comparison with usual polyelectrolytes, the PILs are not soluble in water, but generally in polar organic solvents only [107]. Most PILs exhibit an amorphous solid character or gel-like properties with low transition temperatures. Generally, PILs can be divided with regards to the structure having either (i) cationic or (ii) anionic moieties in the polymer backbone. The most common and most studied PILs belong to the first category, with a variety of mobile anions. The most popular cationic monomers are based on vinylimidazolium [107,109]. The anionic-type PILs are quite rare and thus less reported due to the difficult synthesis [110]. Recently, two reviews on PILs synthesis [108] and properties [111] were published.

### 3.1. Structures of PILs

The term PIL was first applied to the polyelectrolyte compounds synthesized by radical polymerization of ionic liquids with unsaturated bonds [112,113,114,115]. This term later spread in the community of groups working on ILs polymerization and was facilitated by works of Mecerreyes et al. [116,117,118]. Eventually, the term ‘PIL’ also started to be associated with the polymers post-modified chemically to mimic the structure of polymerized ILs [119,120,121,122,123,124]. Currently, the term poly(ionic liquids) refers to several variations, such as “polymerized ionic liquids”, “polymeric ionic liquids”, and “polyionic liquids”, that are used interchangeably, and it is widely applied to the broad range of polymeric materials, comprising a similar structure of repeated electrolyte units bound to a shared polymer backbone. In this work, the term PIL is referring to both polymerized ILs and post-synthesis modified polymers to resemble them. Recently, several contributions have reviewed approaches for PIL synthesis [125,126].

Independent of the synthetic approach, the typical chemical structures are based on several distinct features: Ionic composition, monomer unit homogeneity, and architecture, which, however, are closely intertwined [127,128]. A schematic overview of possible structures is given in Figure 4. The ionic composition differentiates cationic and anionic PILs, containing positively and negatively charged ionic groups bound to the polymer backbone, respectively, as well as polyampholytes that carry both charges simultaneously [129]. Polyampholytes are further subdivided into polyzwitterions and co-polymers. While polyzwitterions have both negative and positive charges bound to the backbone in one pendant [130,131,132,133,134], co-polymers have them bound to the backbone as separate pendants [135,136].

The monomer unit homogeneity provides further differentiation between a single group of homopolymer cationic/anionic PILs and polyampholytes, representing the other group of co-polymers [129]. In addition, co-polymers encompass PILs that are built from both IL-like monomers and neutral monomers [125,137,138,139,140]. PILs diversity is further enriched by the way of their monomer assembly, or more precisely, by their architecture (Figure 4). The major architecture types of PILs are linear (a) [141,142,143], branched and/or hyperbranched (b) [132], and cross-linked (c) [140], where the branched are further subdivided into dendrimer (d) [144], star (e) [145,146], and cyclic (f) [147]. This variability in charge carrying group configurations and polymer architecture not only highlights the creativity of synthetic chemists, but also underscores the versatility of PILs with inexhaustible possibilities for their performance fine-tuning.

### 3.2. Synthesis of PILs

Susceptibility of SILMs to lose ILs during process operation was the main motivation for the development of alternatives, combining the stability of polymers with highly selective functional ILs. The most straightforward approach to obtain PILs is to polymerize suitable ILs into functional polymers via chain growth or step-growth polymerization mechanisms. However, existing neutral polymers can also be functionalized using post-synthetic modifications, which will be discussed later. 

#### 3.2.1. Polymerization of IL Monomers 

Chain growth polymerization (i) requires an IL monomer with an unsaturated or cyclic chemical structure that can be activated during the polymerization step (Figure 5). Such a bond is often synthetically introduced into IL monomers (ILM) through the incorporation of methacrylic and methacrylamide [112,148,149,150], vinyl and styrene [116,151,152,153,154,155], or norbornene [156,157,158] or epoxy moieties. The sheer number of these reports demonstrates the applicability of this approach and underlines its advantages. These mainly include the usage of various synthesis media/solvents, toleration of the reaction mixture impurities, and flexible polymerization approaches [107,127].

Initial reports of PILs synthesis emerged at the beginning of the 2000s and mostly employed conventional free radical polymerization (FRP) [159,160]. While allowing a straightforward preparation of PILs from monomers, it builds from pre-existing double bonds in methacrylate [149,161], vinyl- [155,161,162,163,164], and styrene [154] building blocks. In this case, the synthetic approach would include at least three steps to prepare a PIL with a halide anion: (i) Halide incorporation; (ii) quaternization; (iii) polymerization in the presence of an initiator (i.e., AIBN). The halide anion of PIL can subsequently be altered via an anion-exchange reaction. Interestingly, this approach enables facile incorporation of amino acid anions in PILs promoting the interaction with CO_2_ [155]. Alternatively, the neutralization reaction between tetrafluoroborate (HBF_4_) acid can yield an *N*-vinylimidazolium-derived IL monomer in a smaller number of steps, as no halide incorporation and ion exchange is required [165]. However, the stability of [BF_4_]^-^ is compromised in contact with atmospheric moisture at temperatures above 50 °C during the polymerization, which may result in the emission of the HF acid [166]. The FRP technique offers a major advantage in the preparation of copolymers from a variety of monomers available on the market [107]. The polymerization can be performed according to a selection of well-established processes, such as emulsion, microemulsion, suspension, and dispersion polymerization [127,167,168,169,170,171]. However, the characterization of synthesized PILs is often obscured by the presence of charged groups. For example, the correct determination of the molecular weight distribution by gel permeation chromatography requires the use of special buffers to prevent PILs from sticking onto the column [111]. Moreover, this is often further aggravated by complex control over the polymer synthesis with well-defined molecular weight distribution and architecture [127].

Alternative polymerization techniques have been successfully implemented to enhance the control over the production of PILs with superior CO_2_ selectivity. The controlled polymerizations often used in PILs synthesis for membrane-based gas separation are reversible-deactivation polymerization, nitroxide mediated polymerization [145,172], and ring-opening metathesis polymerization (ROMP) [137,145]. The reversible-deactivation polymerization technique further differentiates reversible addition-fragmentation chain transfer (RAFT), employing a chain transfer agent (i.e., thiocarbonyl) [150,151,173,174,175], and atom transfer radical polymerization (ATRP), employing initiator, catalyst, and ligand [113,176].

Step growth (S-G) polymerization (ii) allows synthesizing PILs in the absence of initiator and termination reactions. Two variations, referred to as polycondensation and addition polymerization, differ from each other based on a formation of a small by-product during the combination of two monomers and the rearrangement of an unsaturated bond into a saturated bond upon the reaction with another molecule, respectively. The majority of the PILs synthesized by a polycondensation employ polyurethane (PU) and polyepoxide S-G chemistry [145,161,177]. A broad variety of PU-based PILs has been prepared to investigate the influence of different ionic diols as cations and counterions on the CO_2_ selectivity [178,179]. Further research on their application in CO_2_/CH_4_ separation demonstrated the possibility of PU-based PILs to overcome the CO_2_ sorption capacity in comparison to linear PILs [121]. Furthermore, several hybrid urethane-imide PILs showed enhanced CO_2_ sorption capacity in the CO_2_ physisorption experiments [180]. Various attempts successfully yielded PILs incorporating polyepoxide building blocks [181,182,183,184,185]. Most of the research uniting PILs and epoxides focuses on PILs usage as catalysts for the fixation of CO_2_ by cycloaddition to form cyclic carbonates [186]. Nevertheless, epoxide polymerization into S-G PILs produced facilitated transport membranes with amino groups as fixed-carrier sites [145,187,188,189].

Polyaddition yields a polymer without a by-product and can proceed with or without an initiator (i.e., 2,2-azobisisobutyronitrile (AIBN), 2,2-dimethoxy-2-phenylacetophenone (DMPA), etc.) and also when UV initiated. In this case, the ionic groups can be situated in the main chain as well as introduced through post-synthetic modification. While polyaddition is less often applied for the preparation of PILs for membrane-based separation, the group of Prof. Drockenmuller has been investigating the possible synthesis routes, properties, and their further applications of 1,2,3-triazole-based PILs with triazolium cation in the main chain [190,191,192,193,194,195]. Suckow et al. synthesized PILs from di-tertiary amines and alkyl dihalides in the attempts to prepare unidirectional polymers via polyaddition [196]. Nevertheless, triazole-based PILs have found little application in the form of gas separation membranes, although their selectivity was comparable to, or even exceeding the market standard for membrane-based CO_2_ capture [197]. 

While generally polymerization of IL monomers offers a flexible and versatile approach for PILs synthesis, this approach requires significant resources in polymer synthesis as monomers often have to be designed to provide CO_2_ separation efficiency. Upon the polymerization, the mechanical properties are then determined by the synthetic assembly of the monomers and their interactions, demanding special attention. This process can be further facilitated by employing modelling to physicochemical intra- and intermolecular interactions. 

#### 3.2.2. Post-Functionalization of Existing Polymers 

Alternatively, PILs can be prepared from existing polymers available on the market or custom-made polymers by their post-functionalization. These polymer precursors are attributed to three different groups based on the presence of a charged group in their backbone or as side groups: (a) Neutral with no chargeable groups (i.e., chitosan (CHI), poly(ethyleneimide) (PEI), poly(4-vinylpyridine) (P4VP), poly(*N*,*N*-dimethyl aminoethylmethacrylate) (PDMAEMA)), (b) anionic polyelectrolytes with anions (i.e., sodium polyacrylate (PAA), sodium polystyrene sulphonate (PSS), sodium hyaluronate (HA)), (c) cationic polyelectrolytes with cations poly(diallyldimethylammonium) chloride (PDADMAC), poly(allylamine) hydrochloride (PAH), and poly(4-vinylbenzyltrimethylammonium) chloride (PVBTMAC)) [126,198,199]. The post-functionalization of neutral polymer precursors is most often conducted via one of the four methods depicted in Figure 6: (i) Ionization, (ii) quaternization to obtain cationic PIL, (iii) sulphation to obtain anionic PIL, and (iv) cross-linking.

Ionization is the most straightforward approach to convert neutral polymer precursors containing amino or carboxylic groups into PILs without any additional chemical reactions. For this, the polymer precursors are dissolved in water at concentrations ranging from 10^−2^ to 10^−4^ mol L^−1^ (moles of monomer unit) with subsequent adjustment of pH with hydrochloric acid (acidification) or sodium hydroxide (basification), dependent on the nature of the polymer [200,201,202,203]. While the polymer ionization offers a swift transition to PILs, sometimes it may require precursor modification to incorporate the ionizable groups themselves. This additional step may further inflate the cost of PIL-based membranes and make PILs less attractive in comparison to conventional materials. Nevertheless, its potential becomes apparent for the post-functionalization of polyimides, which are the commercial standard among gas separation membranes [120].

The majority of cationic PILs are synthesized in two steps: Quaternization (includes imidization) followed by anion exchange. The possibility to incorporate quaternary ammonium groups combined with interchangeable anions opens plenty of opportunities for customization [119,124,204]. Even broader PILs customization can be achieved by incorporating other hetero-ions, such as phosphonium, sulphonium, boronium, etc. The choice of quaternary amino group influences the CO_2_ separation performance through basicity, surface energy, polymer chain packing density, rigidity, etc. The basicity of a cationic group determines the kinetics of CO_2_ interactions with the PILs. To increase the chain packing density, PILs often incorporate aromatic groups with substituents that promote hydrophobic interactions and enable π-stacking [202]. Overall the quaternary ammonium groups yield tighter PILs films through an interplay between electrostatic, hydrophobic, and hydrogen-bonding interactions, which can be further enhanced through ionic cross-linking [205,206]. This so-called “ionic gluing” renders them highly selective towards CO_2_ [150,207].

Sulphation affords the incorporation of anionic moieties into the neutral polymer precursor for the post-functionalization of PILs [206,208]. Alternatively, a similar method where the initial presence of amino alcohol groups in the polymer chain enables the reaction between the hydroxyl and sulphur trioxide, is referred to as sulphamation [209,210]. The incorporation of sulphate and sulphonate anions produces highly resilient PILs through the presence of irreversibly cross-linked ion pairs [211,212,213,214]. Similar stability can be achieved by the incorporation of hydroxide, carbonate, phosphate, and phosphonate anions [121,179]. These anions provide thin films that exhibit not only reversible CO_2_ solubility, but also that resist the hydrolysis of anionic groups.

While *N*,*N*-dimethylaminoethanol methacrylate (DMAEMA)-based PIL membrane could be prepared using the chemical-grafting and co-polymerization discussed earlier, the commercial homopolymer simplified the membrane preparation procedure and decreases the costs of the process [215,216]. Cross-linking can be used to reinforce the stability of commercially available DMAEMA homopolymer using p-xylylene dichloride [217]. This concept can be further expanded towards isocyanatoethyl methacrylate copolymers cross-linked via interfacial polymerization with diethylenetriamine if an additional ionisation step is conducted afterwards [218].

Pseudo-PILs from post-functionalized PIM-1 were prepared by reacting tetrazole-modified PIM-1 with amines in the presence of MeOH to activate anionic sites in the polymer and introduce amines as anions and bearing a methylammonium, diisopropylamonium, and *N*,*N*-diisopropylethylamonium cation [123]. With increasing cation size, the BET surface area and the gas permeability progressively decreased, while the selectivity increased, maintaining values close to the 2008 upper bound for the CO_2_/N_2_ gas pair (selectivity 35.5 at a CO_2_ permeability of 817 Barrer) and the O_2_/N_2_ gas pair (selectivity 4.5 at an O_2_ permeability of 102 Barrer) for the *N*,*N*-diisopropylethylamonium cation. A similar approach can be used to convert 1,2,3-triazole-functionalised polysulphones into PILs [219].

The broad range of methods available for PIL synthesis allows a flexible approach to their design and application. While no solution fits all purposes, no specific method can be branded as the best option. The main parameter affecting the choice of the preferred method depends on the purpose of the PIL and the functionalities it should, therefore, incorporate. However, the final choice between the two options should always rely on the final quality of the product PIL, its ease of production, and economic benefit.

### 3.3. Pure Gas Transport 

#### 3.3.1. Gas Permeation

Comparing to SILMs or IL-based membrane blends, the PILs represent the alternative of the self-standing membranes without the need of blending or the need of a supporting structure. On the other hand, there exist issues and drawbacks connected with the PILs membrane fabrication for gas permeation, such as high costs, poor film-formation, low gas permeabilities on a large scale, etc. [220]. Recently, the performance of PILs in gas separation was briefly reviewed in two scientific reports [23,221]. The pure gas permeabilities of PILs discussed further in this section are summarized in Table 5 and compared to the current state of art of gas separation membranes in Figure 7.

##### Influence of backbone

Cationic PILs reported for the preparation of gas separation membranes can be divided into two main categories: (i) The cationic moiety is anchored to the polymer backbone as side groups, and (ii) the cationic moiety is part of the polymer backbone. In the first group, PILs synthesized from ILs with vinyl- [220,222,223], styrene- [159,224], acrylate- [159,225], and acetylene-based [226] polymer backbones were reported. It was found that vinyl-based PILs are less CO_2_ selective in comparison with styrene- and acrylate-based PILs [223]. *N*-hexyl- and disiloxane- substituted vinyl-PILs had a CO_2_ permeability of 69 and 130 Barrer, respectively. Such values were considerably higher in comparison with styrene- and acrylate-PILs. Linear polyvinyl acetate was incorporated into in situ UV cross-linked PIL (c-PIL, 1-vinyl-3-octylimidazolium hexafluorophosphate ([voim][PF_6_])) [220]. The resulting film exhibited improved mechanical properties and increased CO_2_ permeability with increasing c-PIL content. The best combination of separation properties was reached with a mixture of 50 wt% compositions with the CO_2_ permeability of 36.1 Barrer and CO_2_/N_2_ selectivity equal to 59.6. In this PVAc/c-PIL-50 semi-IPN membrane, the CO_2_ permeability increased, and CO_2_/N_2_ selectivity decreased with increasing temperature. PEG-imidazolium-functionalized polyimide exhibited very high CO_2_ permeability with high CO_2_/CH_4_ and CO_2_/N_2_ selectivity [227]. 

Within the second group, polymer backbones containing (i) imidazolium [228,229,230], (ii) benzimidazolium [231,232,233], triazolium [197,234], pyrrolidinium [235], and pyridinium [235] cationic moieties have been studied. Imidazolium [228,229] or imidazolium-polyimide [230] based backbones were synthesized. Nevertheless, the properties excelled neither in gas permeability nor in ideal selectivity. The highest CO_2_ permeability was 5.3, with moderate CO_2_/N_2_ selectivity of 24.1, and this was found for poly(imidazolium)s (imidazolium ionenes) with decyl spacer and [Tf_2_N]^-^ anion. 

There is no clear evidence of the influence of homopolymeric polybenzimidazole structures in comparison to other PILs with cationic moiety in the polymer backbone. The choice of the anion and the *N*-substituents on polybenzimidazole plays a major role in the permselective properties, as discussed later [231,232]. Two different polyimides with benzimidazolium (PI-1 TFSI and PI-2 TFSI) based PILs were turned out to self-standing films; nevertheless, PI-2 TFSI was too brittle to be suitable for gas separation. PI-1 TFSI exhibited moderate CO_2_ permeability of nearly 29 Barrer but with low selectivity of 2.9 and 4.0 for CO_2_/CH_4_ and CO_2_/N_2_, respectively [233].

Jourdain et al. synthesized a series of 1,2,3-Triazoliumbased linear ionic polyurethanes (PUs) membranes using 1,2,3-triazolium-based diol monomer [234]. The membranes prepared from isophorone diisocyanate/dibutyltin dilaurate and dibutyltin dilaurate/tolylene-2,4-diisocyanate exhibited relatively low CO_2_ permeability (2.31 and 2.25 Barrer). The authors concluded that this is because of the restricted mobility of the polymer chains. On the other hand, cross-linked polyether-based 1,2,3-triazolium membranes had CO_2_ permeability between 59.2 to 113.2 Barrer with CO_2_/N_2_ selectivity close to Robeson’s upper bound [197]. It is most probably due to the combination of flexible PTMEG-based polymer backbones and high affinity of CO_2_ with the polar ether and 1,2,3-triazolium moieties. On the other hand, a copolymer of pyridinium and aromatic polyether exhibited relatively low CO_2_ permeabilities while N_2_ and CH_4_ were on the resolution limit of the measurement set-up [235]. It was found that [MeSO_4_]^-^ based PIL has CO_2_/CH_4_ selectivity of 46 and CO_2_/N_2_ selectivity > 137.

##### Influence of cation type and post-functionalization

The majority of the PILs reported for membrane fabrication are based on the imidazolium moiety. Furthermore, there is consistent agreement that bulkier and longer substituents leads generally to higher gas permeability, while selectivities are either slightly reduced or almost unaffected. For instance, Bara et al. showed that styrene- and acrylate-based imidazolium PILMs can dissolve about twice as much CO_2_ as their liquid analogues [159]. In addition, CO_2_ permeability was reported to increase as a longer n-alkyl substituent was used, while CO_2_/N_2_ separation performance was relatively unaffected as the CO_2_ permeability increased. These results were compared with vinyl-based imidazolium PILs with six different functional cationic substituents [223]. The same research group revealed that introducing polar substituents with comparable length to n-alkyl substituents significantly enhanced the CO_2_/CH_4_ and CO_2_/N_2_ selectivity [236]. Horne et al. showed that imidazolium PILs with branched and cyclic functionalities exhibited around 20% higher CO_2_/N_2_ and CO_2_/CH_4_ selectivities, while gas permeability was more than 50% lower [224]. In accordance with previous studies, n-alkyl groups exhibited increasing CO_2_ permeability with an increasing number of carbons in the pendant group. Shaligram et al. found that the bulkier substituent (such as pyrenyl and anthryl) of benzimidazolium resulted in higher membrane permeability, corresponding to experimentally proved worse chain packing [232]. Vinyl-based polymerized ILs with the [DCA] anion had lower CO_2_ and N_2_ permeabilities in comparison with free RTILs, and lower CO_2_/N_2_ selectivity too. Increasing the length of the alkyl chain on the imidazolinium ring increased the CO_2_ permeability, but decreased CO_2_/N_2_ selectivity [222]. Replacing the bromine group for 1-methylimidazole on the polyacetylene chain of the membranes caused a remarkable increase of the CO_2_/N_2_ selectivity, but it decreased the CO_2_ permeability. The drop in permeability originated from a decrease of both solubility and diffusivity due to stronger intermolecular interactions by the polar imidazolium moiety, while the increase in permselectivity was attributed to the favorable interaction between the CO_2_ molecules and the imidazolium salt [226]. Comparing the works of Kumbharkar et al. [231] and Shaligram et al. [232], it is evident that the bulkiness of substituent on polybenzimidazolium PIL influences both the gas permeability and the ideal selectivity. For instance, substituting one *N*-methyl group to pyrenyl and the second to 4-*tert*-butylbezyl on a PBI PIL with a [Tf_2_N]^-^ anion, resulted in an increase of CO_2_ permeability and CO_2_/N_2_ ideal selectivity from 18 to 36 Barrer and from 7.9 to 20, respectively. 

##### Influence of counter anion

Although the majority of the PILs for membrane fabrication contain the [Tf_2_N]^-^ counter anion [159,197,223,224,225,226,229,231,232,233,234,235,236,237], also other anions such as [BF_4_]^-^ [231,232,237], [Br]^-^ [226,227,228], [DCA]^-^ [222], [I]^-^ [197], [MeSO_4_]^-^ [237], [PF_6_]^-^ [220] and [TfAc]^-^ [226] have been reported. Independent on the nature of the PIL backbone and on the cation, most of the studies concluded that with the increasing bulkiness of counter anion, the permeability of gases increases, usually accompanied by a decrease of CO_2_/gas selectivity. The film-forming ability was studied in detail for benzimidazole-based cation [TMPBI-BuI] with different anions [BF_4_] and [Tf_2_N] [231]. The latter anion led to a more than 3 times higher CO_2_ permeability, without a change in permselectivity toward CH_4_ and N_2_. This was attributed to a lower diffusive resistance due to a looser chain packing, evidenced by the higher d-spacing. Substitution of [Br]^-^ for [Tf_2_N]^-^ resulted in a 20 times higher CO_2_ diffusivity, while solubility increased only slightly [228]. Interestingly, the influence on H_2_ permeability was less profound, and H_2_/CO_2_ selectivity dropped dramatically from 6.5 to 1, indicating a transition from size selectivity to solubility selectivity. PBI based PILs with different anions were compared [232]. The permeability for all gases increases with increasing size of anion as [Br]^-^ (*V_w_* = 16.0 cm^3^ mol^−1^) < [BF_4_]^−^ (*V_w_* = 30.1 cm^3^ mol^−1^) < [Tf_2_N]^-^ (*V_w_* = 88.5 cm^3^ mol^−1^). The authors concluded that [Tf_2_N]^-^ has a better ability to enhance the permeability of the PIL while the [BF_4_]^-^ anion contributes to elevating the selectivity for CO_2_. With the increasing bulkiness of counter anion in the order of Tf_2_N^-^ > TfAc^−^ > Br^−^, the polymer chain packing of poly(diphenylacetylenes)s is thought to be looser, consequently, the CO_2_ permeability increased while the CO_2_/N_2_ permselectivity and solubility selectivity decreased [226]. Aromatic polyether copolymer pyridinium based PILs films with different counter anions were reported as dehumidification membranes. PIL-MeSO_4_ had water vapor permeability of 1.84 × 10^5^ Barrer with outstanding H_2_O/gas selectivities (e.g., H_2_O/CO_2_ = 2 × 10^5^), exceeding the separation performance of the majority of the known membrane materials. The high-water permeation was attributed to solubility-driven rather than diffusivity-driven permeation. Among the tested PIL membranes, the permeability of H_2_, CO_2_, and CH_4_ decreased in the order PIL-Tf_2_N > PIL-BF_4_ > PIL-MeSO_4_, suggesting that the anion has a considerable effect on the gas permeability [237]. Comparing the data on different 1-vinyl-3-alkylimidazolium based PILs, it can be concluded that membranes with [DCA] have lower gas permeabilities and higher ideal selectivities in comparison to those with [Tf_2_N] [222,223]. Contrary to all previous references on the influence of the anion, the change from [I]^-^ to [Tf_2_N]^-^ led to a decrease in the permeabilities of CO_2_ and N_2_ in highly cross-linked polyether-based 1,2,3-triazolium PILs. This permeability decline was attributed to the blocking of gas diffusion by the larger [Tf_2_N]^−^ [197].

#### 3.3.2. Sorption properties

Generally, PILs represent a promising material platform in application fields of CO_2_ capture, separation, or detection. The effects of structural variation in PILs, the choice of the cation, anion, backbone, alkyl chain length, porosity, and cross-linking on the CO_2_ absorption and capturing were thoroughly reviewed in several works [107,241,242,243]. Tang et al. reported the CO_2_ sorption capacity of, e.g., ammonium-based PILs, which was 6 to 7.6 times higher than of corresponding monomers [244]. The same research group showed that radically polymerized alkyl imidazolium-based PILs with tetrafluoroborate and hexafluorophosphate anions exhibited both higher sorption and sorption/desorption kinetics comparing to precursor IL. In comparison, the sorption of other gases, such as N_2_ or O_2_, was not detectable [245]. The CO_2_ sorption capacity of PILs is an important factor determining their overall performance. In summary, the cations play a decisive role in CO_2_ solubility in cationic PILs obtained via radical polymerization. Studies reporting the use of PILs synthesized via condensation polymerization and polymer modification for CO_2_ capture and separation are still less frequent compared to those synthesized by radical polymerization [242].

Imidazolium, phosphonium, ammonium, and pyridinium based polycations were worldwide patented as efficient and selective CO_2_ sorbent [246]. The CO_2_ sorption capacity of PILs can be affected by several polymer properties such as the type of cation, anion, porosity, molecular weight, and alkyl chain length [242]. Comparing to conventional RTILs, the cation plays a major role in CO_2_ sorption, whereas the anion plays a secondary role. Nevertheless, no experimental data are available for CO_2_ sorption for anionic PILs, and such systems have not been examined in detail yet [242]. The studies of the effect of different cation types and different side-chain alkyl groups of the cations resulted in the hypothesis that the higher CO_2_ sorption corresponds to the less delocalized and stronger positive charge on the cation [242,244]. Although the cross-linking increases the microvoid volume, an unexpected decrease of the CO_2_ sorption capacity was observed. Fitting of the data with the dual-mode sorption model showed that the Langmuir sorption capacity increased with the degree of crosslinking, whereas the affinity parameter decreased, thus indicating the hindering effect of CO_2_ affinity to PILs [247]. Tang et al. reported that infinite-dilution Henry constants of polymerized imidazolium-based PILs are lower than those of similar RTILs, indicating that higher CO_2_ sorption in this material is more absorption-based rather than adsorption related [248]. For instance, poly[(p-vinylbenzyl) trimethylammonium hexafluorophosphate] P[[VBTMA][PF_6_], adsorbs 77 wt% of CO_2_ with CO_2_/N_2_ sorption selectivity of 70, although elsewhere in the same manuscript, the authors report 2.5 wt% (Figure 7 vs. Figure 12 in Ref. [249]).

Recently, cationic polyureas with tetrafluoroborate anions have been synthesized. It was shown that CO_2_ sorption depends on the type of the cation and the structure of the diisocyanate, with the highest sorption of 24.8 mg g^−1^ at 0 °C and 1 bar [141].

**Table 5 membranes-11-00097-t005:** Overview of the pure gas permeabilities for CO_2_, CH_4_, N_2_, and H_2_ of a series of PILs.

PIL	Ct. ^(a)^	An. ^(b)^	Back. ^(c)^	*p* [bar]	*T* [°C]	Permeability [Barrer] ^(d)^				Ref.
						CO_2_	CH_4_	N_2_	H_2_	
p[1-Methyl-3-[2-[(1-oxo-2-propenyl)oxy]ethyl]-imidazolium][Tf_2_N]	Im	[Tf_2_N]	Acry	2	20	7	0.19	0.23		[159]
p[1 -Butyl-3-[2-[(1-oxo-2-propenyl)oxy]ethyl]-imidazolium][Tf_2_N]	Im	[Tf_2_N]	Acry	2	20	22	0.97	0.71		[159]
p[1-[(4-Ethenylphenyl)methyl]-3-methyl-imidazolium][Tf_2_N]	Im	[Tf_2_N]	Styr	2	20	9.2	0.24	0.29		[159]
p[1-[(4-Ethenylphenyl)methyl]-3-butyl-imidazolium][Tf_2_N]	Im	[Tf_2_N]	Styr	2	20	20	0.91	0.67		[159]
p[1-[(4-Ethenylphenyl)methyl]-3-hexyl-imidazolium][Tf_2_N]	Im	[Tf_2_N]	Styr	2	20	32	2.3	1.4		[159]
p[C6 bis imidazolium][Tf_2_N]	Im	[Tf_2_N]	PIm	2	20	4.4	0.16	0.2		[229]
p[PEG short bis imidazolium][Tf_2_N]	Im	[Tf_2_N]	PIm	2	20	4	0.12	0.14		[229]
p[PEG long bis imidazolium][Tf_2_N]	Im	[Tf_2_N]	PIm	2	20	3.8	0.13	0.16		[229]
p[1-[(4-Ethenylphenyl)methyl]-3-[2-methoxyethyl]-imidazolium][Tf_2_N]	Im	[Tf_2_N]	Styr	2	22	16	0.48	0.39		[236]
p[1-[(4-Ethenylphenyl)methyl]-3-[2-(2-methoxy-ethoxy)-ethyl]imidazolium][Tf_2_N]	Im	[Tf_2_N]	Styr	2	22	22	0.74	0.5		[236]
p[1-[(4-Ethenylphenyl)methyl]-3-[4-cyano-butyl]-imidazolium][Tf_2_N]	Im	[Tf_2_N]	Styr	2	22	4.1	0.11	0.11		[236]
p[1-[(4-Ethenylphenyl)methyl]-3-[6-cyanohexyl]-imidazolium][Tf_2_N]	Im	[Tf_2_N]	Styr	2	22	8.2	0.28	0.21		[236]
p[poly(imidazolium)s (imidazolium ionenes)with decyl ][Br]	Im	[Br]	PIm	4	RT	0.13			0.84	[228]
p[poly(imidazolium)s (imidazolium ionenes)with decyl ][Tf_2_N]	Im	[Tf_2_N]	PIm	2	RT	5.3	0.26	0.22	5.3	[228]
p[veIm][DCA]	Im	[DCA]	V	10	35	0.09				[222]
p[vbIm][DCA]	Im	[DCA]	V	1	35	4.24		0.16		[222]
p[vhIm][DCA]	Im	[DCA]	V	1	35	33.5		1.6		[222]
p[vmIm][Tf_2_N]	Im	[Tf_2_N]	V	2	RT	4.8	0.1	0.2		[223]
p[vhIm][Tf_2_N]	Im	[Tf_2_N]	V	2	RT	69	7.0	4.1		[223]
p[1-Vinyl-3-[2-(2-methoxy-ethoxy)-ethyl]imidazolium][Tf_2_N]	Im	[Tf_2_N]	V	2	RT	14	0.4	0.4		[223]
p[1-Vinyl-3-[2-[2-(2-methoxy-ethoxy)-ethoxy]-ethyl]]- imidazolium][Tf_2_N]	Im	[Tf_2_N]	V	2	RT	26	1.0	0.8		[223]
p[1-Vinyl-3-(3,3,4,4,5,5,6,6,6-Nonafluorohexyl)-imidazolium][Tf_2_N]	Im	[Tf_2_N]	V	2	RT	69	4.9	6.3		[223]
p[1-Vinyl-3-(1,1,3,3,3-pentamethyl-disiloxanylmethyl)-imidazolium][Tf_2_N]	Im	[Tf_2_N]	V	2	RT	130	14.9	9.3		[223]
p[1-[p-(2-(3-methylimidazolium)ethoxy)]phenyl]-2-[p-(trimethylsilyl)phenyl]acetylen][Br]	Im	[Br]	Ac	1	25	11		0.25		[226]
p[1-[p-(2-(3-methylimidazolium)ethoxy)]phenyl]-2-[p-(trimethylsilyl)phenyl]acetylen][Tf_2_N]	Im	[Tf_2_N]	Ac	1	25	53		1.7		[226]
p[1-[p-(3-methylimidazolium)ethoxy)phenyl]-2-phenylacetylene][Tf_2_N]	Im	[Tf_2_N]	Ac	1	25	8.6		0.25		[226]
p[1-[p-(2-(3-methylimidazolium)ethoxy)]phenyl]-2-[p-(trimethylsilyl)phenyl]acetylen][TfAc]	Im	[TfAc]	Ac	1	25	23		0.67		[226]
p[1-[p-(3-methylimidazolium)ethoxy)phenyl]-2-phenylacetylene][TfAc]	Im	[TfAc]	Ac	1	25	1.5		0.04		[226]
p[diallyldimethylammonium][Tf_2_N]	Pyrr	[Tf_2_N]	Pyrr	1	21	5.09	0.18	0.23		[235]
[TMPBI-BuI][BF_4_]	BIm	[BF_4_]	PBI	19	35	4.78	0.30	0.61		[231]
[TMPBI-BuI][Tf_2_N]	BIm	[Tf_2_N]	PBI	19	35	17.9	1.24	2.27		[231]
pMEBIm[Tf_2_N]	Im	[Tf_2_N]	Acry	1	25	22.8	0.94	0.8		[225]
pMEBIm[Tf_2_N]	Im	[Tf_2_N]	Acry	1	35	33.9	1.99	1.41		[225]
pMEBIm[Tf_2_N]	Im	[Tf_2_N]	Acry	1	45	46.5	3.14	2.26		[225]
pMMA-b-MUBIm[Tf_2_N]	Im	[Tf_2_N]	Acry	1	25	25.8	1.73	0.98		[225]
pMMA-b-MUBIm[Tf_2_N]	Im	[Tf_2_N]	Acry	1	35	33.7	2.73	1.64		[225]
pMMA-b-MUBIm[Tf_2_N]	Im	[Tf_2_N]	Acry	1	45	47.5	4.58	2.67		[225]
p[1-[(4-Ethenylphenyl)methyl]-3-isopropyl-imidazolium][Tf_2_N]	Im	[Tf_2_N]	Styr	3	20	10.4	0.35	0.33		[224]
p[1-[(4-Ethenylphenyl)methyl]-3-sec-butyl-imidazolium][Tf_2_N]	Im	[Tf_2_N]	Styr	3	20	13.6	0.55	0.39		[224]
p[1-[(4-Ethenylphenyl)methyl]-3-methylcyclopropyl-imidazolium][Tf_2_N]	Im	[Tf_2_N]	Styr	3	20	7.94	0.32	0.24		[224]
p[1-[(4-Ethenylphenyl)methyl]-3-cyclopentyl-imidazolium][Tf_2_N]	Im	[Tf_2_N]	Styr	3	20	6.65	0.25	0.19		[224]
p[DPyDBzPBI-BuI][BF_4_]	BIm	[BF_4_]	PBI	20	35	30	0.92	1.32	29.8	[232]
p[DAnDBzPBI-BuI][BF_4_]	BIm	[BF_4_]	PBI	20	35	25.1	0.75	0.92	23.3	[232]
p[DPyDBzPBI-BuI][Br]	BIm	[Br]	PBI	20	35	23.9	0.82	1.11	24.8	[232]
p[DAnDBzPBI-BuI][Br]	BIm	[Br]	PBI	20	35	15	0.49	0.62	15.3	[232]
p[DPyDBzPBI-BuI][Tf_2_N]	BIm	[Tf_2_N]	PBI	20	35	36.2	1.25	1.81	36.5	[232]
p[DAnDBzPBI-BuI][Tf_2_N]	BIm	[Tf_2_N]	PBI	20	35	30.9	1.04	1.43	28.7	[232]
p[6FDA-diethyl-Benzimidazolium][Tf_2_N]	BIm	[Tf_2_N]	PBI	1	20	28.9	7.25	7.23		[233]
p[0.5 cross-linked polyether-based 1,2,3-triazolium][I]	Tr	[I]	PEt		RT	113.2		6		[197]
p[1 cross-linked polyether-based 1,2,3-triazolium][I]	Tr	[I]	PEt		RT	86		2.1		[197]
p[2 cross-linked polyether-based 1,2,3-triazolium][I]	Tr	[I]	PEt		RT	96.5		2.7		[197]
p[1 cross-linked polyether-based 1,2,3-triazolium][Tf_2_N]	Tr	[Tf_2_N]	PEt		RT	59.2		1.2		[197]
p[0.2-PEG-Im-PI][Br]	Im	[Br]	PI	2	30	464	17.4	24.1		[227]
p[0.5-PEG-Im-PI][Br]	Im	[Br]	PI	2	30	177	4.72	5.87		[227]
p[0.8-PEG-Im-PI][Br]	Im	[Br]	PI	2	30	72	1.71	1.94		[227]
p[0.5-Butyl-Im-PI][Br]	Im	[Br]	PI	2	30	219	7.84	11		[227]
p[1,2,3-triazolium-based linear ionic polyurethane (Isophorone diisocyanate)][Tf_2_N]	Tr	[Tf_2_N]	PU	2	20	2.31				[234]
p[1,2,3-triazolium-based linear ionic polyurethane (toluene diisocyanate)][Tf_2_N]	Tr	[Tf_2_N]	PU	2	20	2.25				[234]
p[voim]/PVA IPN 50%[PF_6_]	Im	[PF_6_]	V	1.2	20	18.43		0.26		[220]
p[voim]/PVA IPN 50%[PF_6_]	Im	[PF_6_]	V	1.2	30	29.47		0.46		[220]
p[voim]/PVA IPN 50%[PF_6_]	Im	[PF_6_]	V	1.2	40	38.86		0.74		[220]
p[voim]/PVA IPN 50%[PF_6_]	Im	[PF_6_]	V	1.2	50	41.72		0.83		[220]
p[voim]/PVA IPN 50%[PF_6_]	Im	[PF_6_]	V	1.2	60	42.72		0.89		[220]
p[voim]/PVA IPN 50%[PF_6_]	Im	[PF_6_]	V	1.2	70	43.88		0.92		[220]
p[Polyimide-imidazolium][Tf_2_N]	Im	[Tf_2_N]]	PI	3	RT	0.87	0.028	0.027	1.62	[230]
p[Aromatic polyether copolymer pyridinium ][BF_4_]	Pyr	[BF_4_]	Ar-PEt		30	7.2	0.099	<0.1	13.5	[237]
p[Aromatic polyether copolymer pyridinium ][MeSO_4_]	Pyr	[MeSO_4_]	Ar-PEt		30	0.9	0.05	<0.1	4	[237]
p[Aromatic polyether copolymer pyridinium ][Tf_2_N]	Pyr	[Tf_2_N]	Ar-PEt		30	13.7	0.3	<0.1	13.8	[237]

^(a)^ Ct. = Main type of cation: BIm = Benzimidazolium, Im = Imidazolium, Pyr = Pyridinium, Pyrr = Pyrrolidinium, Tr = Triazolium; ^(b)^ An. = Anion type; ^(c)^ Back. = Polymer Backbone nature: Ac = Acetylene, Acry = Acrylate, Ar-PEt = Aromatic polyether, PBI = Polybenzimidazole, PEt = Polyether, PU = Polyurethane, PI = Polyimide, PIm = poly-Imidazole, Pyrr = Pyrrolidinium, Styr = Styrene, V = Vinyl. ^(d)^ 1 Barrer = 10^−10^ cm^3^_STP_ cm cm^−2^ s^−1^ cmHg^−1^ = 3.348 × 10^−16^ mol m m^−2^ s^−1^ Pa^−1^.

### 3.4. Mixed Gas Transport

Simons et al. reported on mixed gas separation properties of poly(1-[(4-ethenyl phenyl)methyl]-3-alkylimidazolium) bis-(trifluoromethane)sulfonamide) (alkyl = Methyl, n-Butyl, n-Hexyl) for CO_2_/CH_4_ (50/50 vol.%)) up to 40 bar and over a temperature range from 283 to 313 K [250]. They found that the permeability of CO_2_ increased more than 60% over a pressure range of 40 bar; furthermore, both CO_2_ and CH_4_ permeabilities increased with increasing temperatures. Although the plasticizing effect of the absorbed CO_2_ caused an increase in CH_4_ permeability with increasing feed pressure, the permeability of CO_2_ was not suppressed by the presence of CH_4_. Moreover, the short side chains suppressed the plasticizing effect of CO_2_ up to 10 bar of CO_2_ partial pressure. Nikolaeva et al. [154] prepared thin composite PILMs with post-modification of commercial polyvinyl benzyl chloride (PVBC). The membrane with the higher positive charge on the cation species (hydroxyethyl-dimethylammonium > trimethylammonium > pyrrolidinium) exhibited higher CO_2_/N_2_ mixed-gas selectivity. The PILs with the highest CO_2_/N_2_ selectivity (41.6) P[VBHEDMA][Tf_2_N] was further tested in different feed pressures under dry and humid (90% of RH) conditions. While selectivity did not change significantly with pressure, the presence of water led to an increase in selectivity to nearly 50. Butylimidazolium—PMMA based homopolymers and block-copolymer PILs exhibited a linear increase in CO_2_ and CH_4_ permeabilities with feed humidity, while the effect on the gas selectivity was negligible [225]. Recently, PIL-MeSO_4_ membranes were used for the separation of a mixture composed of 3% of H_2_O, 24.25% CO_2_, and 72.75% H_2_ at 30, 80, and 120 °C [237]. Up to two orders of magnitude drop in H_2_0/gas separation factors with respect to the ideal selectivities was observed. The authors concluded that the hydrophilic PIL-MeSO_4_ membrane is plasticized, increasing the CO_2_ permeability 81 times, and the H_2_ permeability 20 times. The H_2_O/gas separation factors further decreased with increasing temperature, accompanied by an increase in gas permeability. The decrease of the H_2_O/CO_2_ separation factor was less pronounced compared to that of the H_2_O/H_2_ due to significant CO_2_ solubility reduction with temperature compared to H_2._

## 4. Polymer/IL Blends; Ion Gels

### 4.1. Introduction

One of the main limitations of supported ionic liquid membranes is the possible loss of the IL when the transmembrane pressure difference exceeds the capillary forces on the IL inside the pores [29,43]. Higher stability requires that the IL is embedded in tighter structures, where the liquid is not free to flow. One of the approaches is the creation of a gel by low molecular-weight organic gelators (LMOG) [251]. However, these LMOG-based gels still need a porous support for mechanical stability. Instead, gels consisting of a low concentration of polymer can provide a highly elastic, soft, and mechanically resistant solid. Such gels have a mesh size of 10 to 100 nm in which the mobility is similar to that of the pure ionic liquid and may be formed using ABA triblock copolymers [252] (Figure 8). Self-assembly of the insoluble end-block upon cooling or upon evaporation of the solvent forms the physical cross-links in the polymer network, containing the soluble mid-block and the ionic liquid. A second possibility to create ion-gel membranes consists of soaking an existing membrane in an excess of the ionic liquid [253], and the amount of IL absorbed in equilibrium can be controlled by the temperature at which it is soaked with IL [254].

Besides self-assembling block-copolymers, semi-crystalline homopolymers may be used, in which the crystalline domains form the physical crosslinks. Hong et al. first reported physical gels of [EMIM][BF_4_] in semi-crystalline poly(vinylidene fluoride-co-hexafluoropropylene) fluoroelastomer, p(VDF-HFP), reaching a strongly enhanced CO_2_/N_2_ selectivity (*α* = 60) compared to the neat polymer, at a CO_2_ permeability of 400 Barrer [255]. The groups of Jansen, Izak, and Friess then prepared ionic liquid gels of [EMIM][TFSI] ([TFSI] is also reported as [Tf_2_N]) in p(VDF-HFP) [254,256] and [BMIM][CF_3_SO_3_] in Pebax^®^ poly(ether-*b*-amide) thermoplastic elastomer [257]. These systems allow the transport properties and the mechanical properties of the membranes to be tailored over a wide range by variation of the IL content from neat polymer up to 80%. The physical cross-links responsible for the gelation of the polymer/IL blend are formed by crystalline domains in the semi-crystalline polymers, and by microphase separation and/or crystallization in the thermoplastic elastomers. Numerous membranes have been developed [19], based on a variety of different combinations of IL and crystalline polymers, thermoplastic elastomers, and glassy amorphous polymers (Table 6). This section aims to provide a complete overview of membranes where the RTIL is mixed with a gel-forming polymer.

### 4.2. Conventional Glassy and Crystalline Polymers

Gelatin is the best-known gel-forming polymer for its widespread application as a food ingredient in jams, jelly, pudding, etc. Couto et al. used it as a gelating agent for [EMIM][DCA], [BMIM][DCA], and [BMPyr][DCA] [258]. They thus produced ion-jelly^®^ membranes, which needed porous support to form robust films, and reached lower selectivities than those of the neat ILs, probably due to the presence of defects in the films. Attempts by Lam et al. to use the well-known cellulose triacetate (CTA) doped with [EMIM][BF_4_] and [EMIM][DCA] were more successful and resulted in robust self-standing films with a *T_g_* that decreased from 198 °C for the pure CTA to just below room temperature for the samples with approximately 50% IL [259]. Their N_2_ and CH_4_ permeability coefficients were almost constant over the pressure range from 4 to 11 atm, but that of CO_2_ increased by about 20–50% with pressure in this range, and all gases showed typical dual-mode sorption behavior. Diffusivity increased with increasing IL content, but solubility decreased, and the combined result was a significant increase in CO_2_ permeability, but nearly constant selectivity, nevertheless reaching the 2008 Robeson upper bound. [EMIM][DCA] was also blended with polyvinyl alcohol (PVA) membranes, where it acted as a strong plasticizer, dramatically reducing the crystallinity, the melting point, and the glass transition temperature of the polymer [51]. This turned the membrane from a size-selective polymer with preferential H_2_ permeation to a ‘reverse selective’ polymer with αCO_2_/H_2_ = 7.2 for 53% IL.

#### 4.2.1. Glassy High-Performance Polymers

Glassy polymers stay homogeneous and maintain their integrity if the compatibility of the ionic liquid is sufficient and if it does not plasticize the polymer too much, so that the glass transition temperature remains higher than the membrane operation temperature. Several membranes have been investigated successfully based on glassy polymers which are already used as gas separation membranes in their neat form. Amorphous glassy polyethersulfone/[EMIM][Tf_2_N] blend membranes were prepared and remained homogeneous up to an IL content of 50% [260], whereas the same ionic liquid gave phase separation already at 20% in the polyimide Matrimid^®^5218 [261]. Instead, [BMIM][Tf_2_N] showed considerably higher compatibility with the 6FDA- and 2,3,5,6-tetramethyl-1,4-phenylene diamine-based polyimide (6FDA-TeMPD) [262], where the samples remained homogeneous up to 51% of the IL and showed distinct phase separation at 80% IL. [BMIM][Tf_2_N] proved also fully compatible with polybenzimidazole (PBI) and with the polyimide poly(pyromellitimide-*co*-4,4′-oxydianiline) (PMDA-ODA) [263]. The latter two polymers allowed gas permeation measurements up to 200 °C for successful separation of H_2_ from CO_2_. Ito et al. increased the compatibility of the polymer with the ionic liquid by sulfonation of the polyimide-based on 1,4,5,8-naphthalene-tetracarboxylicdianhydride and bis[4-(3-aminophenoxy)-phenyl]sulfone and using the same Tf_2_N^-^ or BF_6_^-^ anion as in the [BMIM][Tf_2_N] and [BMIM][BF_6_] ionic liquids [264].

#### 4.2.2. PIM/IL Blends

Polymers of intrinsic microporosity (PIMs) form a relatively new class of polymers and represent an extreme example of high-performance polymers. They were introduced by Budd and McKeown in 2004 [265,266], and they owe their exceptional gas permeability to their high fractional free volume, as a result of the rigid contorted polymer backbone, which does not allow them to pack space efficiently. Although the combination with ionic liquids, which are likely to fill part of this free volume and may also plasticize the polymers, is counter-intuitive, some researchers have nevertheless made blends with the archetypal PIM-1 and [C_2_MIM][Tf_2_N], [C_4_MIM][Tf_2_N], and [C_6_MIM][Tf_2_N] [267]. The presence of 5% IL led to a progressive decrease in CO_2_ permeability from 7440 Barrer to 6650, 4590, and 2240 Barrer for the [C_2_MIM], [C_4_MIM], and [C_6_MIM] cation, respectively, along with an increase in CO_2_/N_2_ selectivity from 19 to 25 and CO_2_/CH_4_ selectivity from 11 to 14 for [C_6_MIM][Tf_2_N]. Due to poor compatibility with the polymer, the ILs partially segregated into ellipsoidal domains. The compatibility was increased by copolymerization of the PIM-1 monomers with a PEG functionalized anthracene maleimide comonomer, resulting in a final CO_2_ permeability of 812 Barrer and a CO_2_/N_2_ and CO_2_/CH_4_ selectivity of 30 and 19, respectively, at a [C_6_MIM][Tf_2_N] concentration of 10%. Han et al. [268], doped PIM-1 with the 1-butyl-3-methylimidazolium tetraisothiocyanatocobaltate(II) complex ([BMIM]_2_[Co(NCS)_4_]) as a carrier for facilitated O_2_ transport, comparing its performance with that of tetra(4-chlorophenyl)porphyrin cobalt (T(4-Cl)PPCo). The fractional free volume of the membranes decreased and the density increased with the [BMIM]_2_[Co(NCS)_4_] concentration, indicating that the complex likely occupied the existing free volume elements of PIM-1. Only small amounts were added, which were not able to form a gel, but in the case of [BMIM]_2_[Co(NCS)_4_] it effectively increased the O_2_/N_2_ selectivity from 3.77 for pure PIM-1 to 5.3 for the membrane with 2% of the cobalt complex. In an alternative approach, Guiver et al. covalently connected the ionic liquid-like tetrazole-type structures as lateral substituents to the PIM-1 backbone [123], resulting in a PIL.

#### 4.2.3. Semi-Crystalline Fluoropolymers

Crystalline fluoropolymers are usually barrier materials and serve more commonly as the porous polymer support for SILMs. Their permeability increases quickly when blended with ionic liquids, which reduce the overall crystallinity and increases the amorphous phase and increase the overall flexibility. These were among the first materials to be used as ion gel membranes. Chen et al. show that the PVDF/[EMIM][B(CN)_4_] blend is anisotropic and the correlation between the transport properties and the microstructure can be described quite satisfactorily by the Maxwell equation [269]. In this system, the addition of almost 80% of [EMIM][B(CN)_4_ leads to a four-fold increase in both permeability and diffusivity, whereas the solubility increases just slightly more than 50%. Song et al. perform simulation studies of blends based on PVDF and [BMIM][PF_6_], [BMIM][Tf_2_N], and [BMIM][B(CN)_4_], confirming the microstructure with IL-rich and PVDF-rich domains [270].

The fluoroelastomer poly(vinylidene fluoride-*co*-hexafluoropropylene), p(VDF-HFP) has a lower crystallinity than PVDF and is also soluble in common volatile solvents such as acetone, which facilitates the preparation of membranes by solvent evaporation. Hong et al. first reported physical gels of [EMIM][BF_4_] in p(VDF-HFP) with up to 67% of IL, which yielded a membrane with a CO_2_ permeability of 400 Barrer and a CO_2_/N_2_ selectivity equal to 60, which exceeded the 2008 upper bound [255]. The groups of Jansen, Izak, and Friess reported ionic liquid gels with this polymer and up to 80 wt% of 1-ethyl-3-methylimidazolium bis(trifluoromethylsulfonyl)imide, [EMIM][Tf_2_N] (or [EMIM][TFSI]) [254]. A gradual decrease of the crystallinity and Young’s modulus was responsible for the strong increase in permeability and the gradual transition from size-selective to solubility-selective transport with reverse selectivity for CO_2_/H_2_ [256] (Figure 9). Uchytil et al. found surprisingly high permeability for the same p(VDF-HFP)/[EMIM][Tf_2_N] system and p(VDF-HFP)/[HMIM][Tf_2_N] in a sandwich configuration, with ideal CO_2_/CH_4_ selectivity up to 32 and 59, respectively [271]. A mixture of [EMIM][Tf_2_N] and [HdMIM][Tf_2_N] in p(VDF-HFP) yielded membranes with a temperature-dependence of the gas permeability around the melting temperature of [HdMIM][Tf_2_N] [272]. The p(VDF-HFP)/[EMIM][Tf_2_N] blend with 80% IL shows very high permeability for linear alcohols, which decreases from 20,000 to 7000 Barrer in the order MeOH > EtOH > PrOH > BuOH, whereas the permeability of linear alkanes increases with chain length, from ca. 200 for butane to 900 for heptane [256].

The alcohol vapor permeability is several orders of magnitude higher than that of light gases, and only CO_2_ has a similar permeability as pentane and hexane, but still more than one order of magnitude lower than that of the alcohols. With respect to the neat polymer, the EtOH solubility increases by about one order of magnitude in the membrane with 80% [EMIM][Tf_2_N], while the diffusivity increases over 3 orders of magnitude [273]. Thus, the high permeability of alcohols depends both on the solubility and on the diffusivity. Ethanol sorption and diffusion strongly depend on the vapor activity and the IL content, and the difference with the permanent gases is mostly due to the higher solubility of the vapors [256], which makes these membranes potentially suitable for the selective removal of volatile organic compounds from gas streams [274].

#### 4.2.4. Thermoplastic Elastomers

The thermoplastic elastomers Pebax^®^ and PolyActive^®^ are effective membranes already in their neat form and represent one of the largest groups of membrane materials for the preparation polymer/IL blends. Various groups have studied the effect of ionic liquid on the transport properties of Pebax membranes after the work of Bernardo et al. [257], who studied the effect of [BMIM][CF_3_SO_3_] on the transport properties of Pebax^®^2533 and Pebax^®^1657 up to 80% loading. In Pebax^®^1657, [BMIM][CF_3_SO_3_] reduced the crystallinity, the melting point of the polyamide phase, and Young’s modulus of the films, increasing the permeability. Pebax^®^2533 was much less affected by the IL, which was also observed for Pebax^®^2533 thin-film composite membranes doped with the task-specific ionic liquid (TSIL) triethylenetetramine trifluoroacetate ([TETA][Tfa]) under dry conditions [275]. In contrast, the use of a high relative humidity resulted in an almost three-fold increase in permeability of the membrane with 30% of the TSIL, and a 50% increase in CO_2_/N_2_ mixed gas selectivity. Using [EMIM][EtSO_4_], Bhattacharya and Mandal found that the H_2_S permeability of Pebax^®^2533 increased almost 350%, with as little as 5% of the IL, while the ideal H_2_S/CH_4_ selectivity and CO_2_/CH_4_ selectivity both more than doubled [276].

Sorption analysis in Pebax^®^1657 with [EMIM][BF_4_] showed a significant increase in the solubility of CO_2_, N_2_, and H_2_, whereas CH_4_ showed a remarkable decrease in solubility with increasing IL content [277]. Nevertheless, the CH_4_ permeability strongly increased with increasing IL content, evidently by a strong increase in the diffusion coefficient. With the same blend, thin-film composite membranes on a PVDF support with a PTMSP gutter layer were successfully prepared [278], showing somewhat higher CO_2_/N_2_ and CO_2_/CH_4_ selectivity with mixed gas than with pure gases. Both works reported a strong increase in permeability and a decrease in selectivity as a function of temperature, similar to the system Pebax1657/[BMIM][Tf_2_N] [279], because of a reduction in both diffusion selectivity and solubility selectivity. The faster diffusion was related to the strong decrease in Young’s modulus according to the tensile test results [277]. A similar trend was observed recently for Pebax^®^1657 blended with [BMIM][BF_4_], as confirmed by both tensile tests [280,281] and by AFM force spectroscopy measurements [280]. Li et al. presented a systematic study on the variation of the anion and the alkyl length of the C_n_MIM-based ILs to tailor the microstructure of the blends in Pebax^®^1657 and found the best performance of the blend with 20% [C_6_MIM][Gly], achieving a CO_2_ permeability of 1490 Barrer and a CO_2_/N_2_ ideal selectivity of 95. Pardo et al. studied the separation of difluoromethane (R32) and pentafluoroethane (R125), the constituents of the commercial refrigerant R410A, with four different EMIM-based ILs in Pebax^®^1657 [282]. The blends with 40 wt% [EMIM][SCN] and [EMIM][BF_4_] gave the best R32/R125 selectivities up to 14.5 and 11.0, respectively, 110% and 60% higher than that of the neat polymer. The membranes also maintained good performance over almost a month at high pressure. Mahdavi et al. used Pebax^®^1074, which has a similar polyamide/polyether ratio as Pebax^®^1074, with PA12 instead of PA6 as the hard phase. Addition of 20–80% [BMIM][PF_6_] caused an approximately two-fold increase in CO_2_ and CH_4_ permeability, with a 10% decrease in selectivity. The addition of 2–8% SiO_2_ particles to the membrane with 80% IL, gave a further increase of ca 50% of the permeability in combination with a 5% increase of the CO_2_CH_4_ selectivity, thus approaching the 2008 Robeson upper bound.

PolyActive^TM^ is a similar multi-block copolymer as Pebax^®^, with the main difference that the hard block is polyester and not a polyamide. Klepić et al. presented a study on blends of PolyActive^TM^ with [BMIM][Tf_2_N] and the effect of the copolymer composition and the IL concentration on the morphology, the thermal and mechanical properties [283] and their pure and mixed gas transport properties [284]. In this study, the polyester phase was nearly unaffected by the IL, which reduced the melting temperature and crystallinity of especially the polyether phase, as well as Young’s modulus of the polymers. This was reflected in an increase of the permeability as a function of the IL content, mainly due to an increase in the diffusion coefficient. The CO_2_/N_2_ and CO_2_/CH_4_ mixed gas selectivity were close to the ideal selectivity and were almost independent of the pressure up to 6 bar. The PolyActive^TM^ grade with a combination of high polyether content and a low block length showed the highest CO_2_ permeability.

#### 4.2.5. Di- and Tri-Block Copolymers

Block copolymers with well-defined block lengths may offer better control of the membrane microstructure via self-assembly of the individual blocks in different domains, in comparison with the multiblock thermoplastic elastomers. Yoon et al. [285] reported poly(styrene-*block*-ethylene oxide-*block*-styrene) (SEOS) triblock copolymer membranes with hydrophilic mid-block, doped with the [Tf_2_N^−^], [BF_4_^−^], and [PF_6_^−^] salts of 1-n-hexyl-3-methylimidazolium ([HMIM^+^])-based ionic liquids, and Hwang et al. made ion gel membranes of three 1-alkyl-3-methylimidazolium tricyanomethanide ILs ([RMIM][TCM]) with 5% SEOS [286]. Both used a porous polyamide as a mechanical support, while Gu et al. doped similar poly(styrene-*b*-ethylene oxide-*b*-styrene) (SOS) triblock copolymer and poly(styrene-*b*-methyl methacrylate-*b*-styrene) (SMS) triblock copolymers with a hydrophobic mid-block with the ionic liquid 1-ethyl-3-methylimidazolium bis(trifluoromethylsulfonyl)amide ([EMIM][TFSA]) [287] and used porous PVDF as a mechanical support. Instead, melt-processed blends of polystyrene-*b*-poly(ethylene oxide) diblock copolymer and polystyrene-*b*-poly(ethylene oxide)-*b*-polystyrene triblock copolymer (SO/SOS) yielded elastic self-standing films upon soaking with up to 94% of [EMIM][TFSI] [288]. With their polyethylene oxide mid- or end-blocks, these polymers were designed for CO_2_ separation from N_2_ or CH_4_, and they successfully improved the permeability and/or selectivity, reaching values close to or exceeding the 2008 Robeson upper bound. Instead, the SMS triblock copolymer was able to form a gel with the [EMIM][TFSA], but did not improve the permeability and selectivity substantially with respect to the pure IL. The pentablock copolymer poly[*tert*-butylstyrene-*b*-(ethylene-*alt*-propylene)-*b*-(styrene-*co*-styrenesulfonate)-*b*-(ethylene-*alt*-propylene)-*b*-*tert*-butylstyrene] (Nexar^®^), reported by Dai et al. [289], was functionalized with sulfonate groups in the mid-block to increase its polarity and its affinity for CO_2_. This polymer also behaves as a thermoplastic elastomer and successfully formed a self-standing CO_2_/N_2_ separation membrane. The addition of [BMIM][BF_4_] greatly enhanced its permeability, in particular in the presence of a humidified feed stream, where it reached a CO_2_ permeability up to 194 Barrer and an impressive CO_2_/N_2_ selectivity equal to 128.

### 4.3. Chemically Cross-Linked Gels

Stable gels can be prepared by chemical cross-linking of different reactive monomers, bearing double bonds or epoxy groups, in the presence of an ionic liquid. The multifunctional monomer creates the polymer network, embedding the ionic liquid.

#### 4.3.1. Acrylic Networks

Kasahara et al. first prepared a series of cross-linked ionic liquid gels for CO_2_/N_2_ separation by free radical polymerization of hydrophilic acrylic or vinyl monomers, using ethylene glycol dimethacrylate as a crosslinker and tetrabutylphosphonium type ionic liquids based on four amino acids (Gly, Ser, Lys, Pro) [290]. Further improvement of the mechanical properties was obtained by the formation of so-called double network gels, one phase based on the rigid polyelectrolyte poly(2-acrylamido-2-methyl-1-propanesulfonic acid) (PAMPS), and the other phase formed by ion gels polyvinylpyrrolidone (PVP) and polydimethylacrylamide (PDMAAm) with tetrabutylphosphonium prolinate ([P_4444_][Pro]) and triethyl(pentyl)phosphonium prolinate ([P_2225_][Pro]) as the aminoacid ionic liquids [291]. The same authors reported impressive CO_2_/N_2_ separation performance for very low CO_2_ concentrations (500–1000 ppm) at high relative humidity. The system showed facilitated transport of CO_2_, with a permeability of 52,000 Barrer and a CO_2_/N_2_ selectivity of 8100 [292] for a double network with the IL [P_4444_][Pro] at 70% RH. These values are even far above the most recent upper bound for polymers of intrinsic microporosity [240]. Similar double network membranes based on poly(methacryloylamino propyl trimethylammonium chloride) (PMAPTAC) with N,N′-methylenebis(acrylamide) (MBAA) crosslinker and poly(dimethylacrylamide) (PDMAAm) with [P_222(1O1)_][Inda] also reached extremely good performance, with CO_2_ permeability up to 20,000 Barrer and CO_2_/N_2_ selectivity of ca. 950 at low CO_2_ partial presure [293].

#### 4.3.2. Epoxy-Based Networks

Epoxy-amine ion gel membranes trap RTILs in a cross-linked epoxy resin network based on a multifunctional amine and an IL functionalized with two or more epoxy groups [145]. The use of an excess of amino groups allows for the fixed-site-carrier facilitated transport of CO_2_ via their free amino groups. Both for CO_2_/N_2_ mixture separation Figure 10, [188]) and CO_2_/CH_4_ mixture separation [189], the presence of humidity enables the facilitated transport mechanism.

The creation of an inorganic/organic network provided a remarkable increase in the toughness of ion gel membranes based on polydimethylacrylamine (PDMAAm) and 80 wt% of [EMIM][Tf_2_N], compared to the mechanical properties of the purely organic network of PDMAAm with [EMIM][Tf_2_N] [294]. With different ILs, Young’s modulus of the inorganic/organic network is approximately twice as high as that of the neat polymer network, and the fracture stress may increase up to almost 10 times [295]. These ion gel membranes have also a remarkably high permeability, which is in some cases even higher than that of analogous SILMs. Commercial Elvaloy4170, epoxide-functional poly(ethylene methyl methacrylate) copolymer was reported to show increasing permeability and CO_2_/CH_4_ selectivity with increasing [EMIM][Tf_2_N] concentration from 0 to 40 wt%, but despite the available epoxy groups, the authors did not cross-link this polymer [296].

#### 4.3.3. Cross-Linked Polyether Networks

Analogous to the Pebax^®^ physically crosslinked poly(ether-amide) block copolymers, chemically cross-linked polyether-based ion gels were reported as well. Among twelve different ILs, the terminal substituent of the cation was found to determine most the extent of improvement of the gas permeability, and shorter substituents proved most effective because they were able to guarantee the highest charge density [297] in ethoxylated trimethylolpropane triacrylate-based ion gels. On the other hand, when varying the polymer network structure, a lower cross-link density tends to yield a higher permeability because it allows the incorporation of more [EMIM][Tf_2_N]. A lower cross-link density also improves the tensile properties [298]. For successful membrane preparation, the compatibility of the IL is of particular interest, and in a series of cross-linked PEO ion gels, 1,3-substituted imidazolium-based ILs with an aromatic cation and acidic proton and with a low basicity anion (such as [Tf_2_N]) show good miscibility and high thermal stability. Copolymerization with a polysiloxane reduces the compatibility of the network polymer with the ILs, although it improves the permeability [299].

### 4.4. Facilitated Transport Polymer/IL Gels

Facilitated transport membranes provide an additional transport mechanism to the IL-based membranes. Besides the standard solution-diffusion mechanism, a reaction takes place between the membrane material and at least one of the permeating species. The reactive sites selectively promote solubility or diffusivity, or both, and maybe part of the polymer network (fixed site carrier) or the IL phase (mobile carrier).

#### 4.4.1. Fixed-Site Carriers

Amine groups may interact with or react with carbon dioxide molecules and thus increase their affinity for the membrane. Amine functional groups in the polymer network provide a fixed-site-carrier for facilitated transport of CO_2_, enhancing CO_2_/N_2_ mixture separation [188] and CO_2_/CH_4_ mixture separation [189], in particular in the presence of humidity, which promotes the facilitated transport mechanism.

#### 4.4.2. Mobile Carriers

Amino Acid ILs were found to act as facilitated transport carriers for CO_2_ in a [P_4444_][Pro]-PVP gel film, as deduced from the decrease in permeability with increasing CO_2_ partial pressure [290]. This is a well-known tendency for facilitated transport membranes, where the carrier sites become saturated with increasing CO_2_ partial pressure, analogous to the trend in high-free volume glassy polymer membranes, where the Langmuir sorption sites become saturated. A similar decrease in CO_2_ permeability with increasing pressure suggests a dominant role for facilitated transport of CO_2_ also in Double Network ion-gel membranes with different triethyl(2-methoxymethyl)phosphonium indazole ([P_222(1O1)_][Inda]) concentrations [293]. These amine-functional task-specific ionic liquids (TSILs) form carbamic acid and/or carbamate IL-CO_2_ complexes via a reversible chemical reaction with CO_2_. Ionic liquid metal complexes have been used as a carrier for various gases. A cobalt complex, in the form of [BMIM]_2_[Co(NCS)_4_], served as a carrier for O_2_ in O_2_/N_2_ separation [268], and only 2 wt% of it increased the O_2_/N_2_ selectivity from 3.5 to 5.3, with a loss of O_2_ permeability from 220 Barrer to 110 Barrer. Up to 0.4% of the IL, permeability, and selectivity increase simultaneously in a constant fractional free volume, suggesting the facilitated transport mechanism, but the subsequent loss in permeability and a further increase in selectivity must be attributed mostly to the loss of free volume of the polymer upon addition of the IL. The copper salt [C_4_MIM][CuCl_2_] in Poly([C_4_VIM][Tf_2_N], designed for the selective transport of carbon monoxide, was unexpectedly found to be ineffective for the N_2_/CO separation [300], but increased the size-selectivity of the membranes and had a favorable effect on the H_2_/N_2_ ideal selectivity. Silver salts are known to be active as carriers for facilitated transport of olefins in olefin/paraffin separation, and this principle was used by Kasahara et al. for propylene/propane separation with solutions of [EMIM][Tf_2_N]/AgTf_2_N and [EMIM][BF_4_]/AgBF_4_, and the oligomeric gelator Poly[pyridinium-1,4-diyliminocarbonyl-1,4-phenylene-methylene [301]. However, the low molecular weight gelator 12-hydroxystearic acid in [EMIM][Tf_2_N]/AgTf_2_N yielded more stable gels with higher C_3_H_6_/C_3_H_8_ selectivity, up to 15 at low pressure. An overview of all polymer/IL blend systems is given in Table 6.

**Table 6 membranes-11-00097-t006:** Polymer/IL blends reported in the literature for pure gas permeation and mixed gas separation.

Polymer	Ionic Liquid	Gases ^(a)^	Ref.
Linear polymers/IL blends			
*Various polymers (hydrophilic)*			
Gelatin	[EMIM][DCA], [BMIM][DCA], [BMPyr][DCA]	H_2_, N_2_, O_2_, CO_2_, CH_4_	[258]
Cellulose Triacetate	[EMIM][BF_4_], [EMIM][DCA]	N_2_, CH_4_, CO_2_	[259]
Polyvinylalcohol, PVA	[EMIM][DCA]	H_2_, CO_2_	[51]
*Glassy high-performance polymers*			
Polyethersulfone	[EMIM][Tf_2_N]	CH_4_, CO_2_	[260]
Matrimid^®^5218 polyimide	[EMIM][Tf_2_N]	CH_4_, CO_2_	[261]
6FDA-TeMPD polyimide	[BMIM][Tf_2_N]	H_2_, N_2_, CH_4_, CO, CO_2_	[262]
Polybenzimidazole (PBI)	[BMIM][Tf_2_N]	H_2_, N_2_, CH_4_, CO, CO_2_ (Elevated T)	[263]
Poly(pyromellitimide-co-4,4′-oxydianiline) (PMDA-ODA PI)	[BMIM][Tf_2_N]	H_2_, N_2_, CH_4_, CO, CO_2_ (Elevated T)	[263]
Sulfonated polyimide SPI-[Tf_2_N]	[BMIM][Tf_2_N]	N_2_, CO_2_	[264]
Sulfonated polyimide SPI-[PF_6_]	[BMIM][PF_6_]	N_2_, CO_2_	[264]
*High free volume polymers*			
PIM-1	[C_2_MIM][Tf_2_N], [C_4_MIM][Tf_2_N], [C_6_MIM][Tf_2_N]	O_2_, N_2_, CH_4_, CO_2_	[267]
PIM-1/PEG functionalized anthracene maleimide comonomer	[C_6_MIM][Tf_2_N]	O_2_, N_2_, CH_4_, CO_2_	[267]
PIM-1	[BMIM]_2_[Co(NCS)_4_] (on PAN)	O_2_, N_2_	[268]
Tetrazole-modified PIM-1	[Methylammonium], [Diisopropylamonium], [N,N-diisopropylethylamonium]	O_2_, N_2_, CO_2_	[123]
*Semi-crystalline fluoropolymers*			
PVDF	[EMIM][B(CN)_4_]	H_2_, N_2_, CO_2_ **CO_2_/H_2_, CO_2_/N_2_**	[269]
	[BMIM][PF_6_], [BMIM][Tf_2_N], [BMIM][B(CN)_4_]	CO_2_ (modelling)	[270]
Poly(VDF-HFP)	[EMIM][BF_4_]	**CO_2_/N_2_**	[255]
	[EMIM][TFSI]	He, H_2_, O_2_, N_2_, CH_4_, CO_2_	[254]
	[EMIM][TFSI]	He, H_2_, O_2_, N_2_, CH_4_, CO_2_Butane, pentane, hexane, cyclohexane, heptane, toluene, MeOH, EtOH, PrOH, BuOH, H_2_O	[256]
	[EMIM][Tf_2_N], [HMIM][Tf_2_N]	CH_4_, CO_2_	[271]
	[EMIM][TFSI]/[HdMIM][TFSI]	He, H_2_, O_2_, N_2_, CH_4_, CO_2_ R-(+)-limonene/S-(+)-carvone/EtOH(/H_2_O)	[272]
	[EMIM][Tf_2_N]	EtOH	[273]
	[EMIM][Tf_2_N]	Hexane, isooctane	[274]
*Thermoplastic elastomers*			
Pebax^®^ 2533	[BMIM][CF_3_SO_3_]	He, H_2_, O_2_, N_2_, CH_4_, CO_2_	[257]
	[TETA][Tfa]	**CO_2_/N_2_**	[275]
	[EMIM][EtSO_4_]	H_2_S, CH_4_, CO_2_, Air	[276]
Pebax^®^ 1657	[BMIM][CF_3_SO_3_]	He, H_2_, O_2_, N_2_, CH_4_, CO_2_	[257]
	[EMIM][BF_4_]	H_2_, N_2_, CH_4_, CO_2_	[277]
	[EMIM][BF_4_] (on PVDF/PTMSP HF)	N_2_, CH_4_, CO_2_ **CO_2_/N_2_, CO_2_/CH_4_**	[278]
	[BMIM][BF_4_]	N_2_, CH_4_, CO_2_	[281]
	[BMIM][BF_4_]	He, H_2_, O_2_, N_2_, CH_4_, CO_2,_ **CO_2_/N_2_, CO_2_/CH_4_**	[280]
	[EMIM][Gly], [BMIM][Gly], [HMIM][Gly], [BMIM][Ac], [BMIM][BF_4_]	N_2_, CH_4_, CO_2_ **CO_2_/N_2_, CO_2_/CH_4_**	[302]
	[BMIM][Tf_2_N]	H_2_, N_2_, CH_4_, CO_2_	[279]
	[EMIM][SCN], [EMIM][BF_4_], [EMIM][OTf], [EMIM][Tf_2_N]	CH_2_CF_2_ (R32), CF_3_CHF_2_ (R125) **CH_2_CF_2_/CF_3_CHF_2_** (R410A)	[282]
Pebax^®^ 1074	[BMIM][PF_6_]	CH_4_, CO_2_	[303]
PolyActive™ (4 grades) 1500PEOT77PBT23, 4000PEOT77PBT23, 1000PEOT55PBT45, 4000PEOT50PBT50	[BMIM][Tf_2_N]	He, H_2_, O_2_, N_2_, CH_4_, CO_2_**CO_2_/N_2_, CO_2_/CH_4_**	[284]
*Di- and tri-block copolymers*			
Poly(styrene-*block*-ethylene oxide-*block*-styrene) (SEOS)	[HMIM][Tf_2_N], [HMIM][BF_4_], [HMIM][PF_6_] (on porous PA)	CH_4_, CO_2_	[285]
	[EMIM][TCM], [BMIM][TCM], [HMIM][TCM] (on porous PA)	N_2_, CH_4_, CO_2_	[286]
Poly(styrene-*b*-ethylene oxide-*b*-styrene) (SOS)	[EMIM][TFSA] (on PVDF)	N_2_, CH_4_, CO_2_	[287]
Poly(styrene-*b*-Methyl methacrylate-*b*-styrene) (SMS)	[EMIM][TFSA] (on PVDF porous support)	N_2_, CH_4_, CO_2_ **CO_2_/N_2_, CO_2_/CH_4_**	[287]
Polystyrene-*b*-poly(ethylene oxide) diblock copolymer/polystyrene-*b*-poly(ethylene oxide)-*b*-polystyrene triblock copolymer (SO/SOS)	[EMIM][TFSI]	N_2_, CH_4_, CO_2_	[288]
Poly[*tert*-butylstyrene-*b*-(ethylene-alt-propylene)-*b*-(styrene-co-styrenesulfonate)-*b*-(ethylene-alt-propylene)-*b*-*tert*-butylstyrene] pentablock copolymer (Nexar)	[BMIM][BF_4_]	N_2_, CO_2_ (H_2_O) **CO_2_/N_2_ (H_2_O)**	[289]
*Low molecular weight gelators*			
Aspartame-based, low molecular-weight organic gelator	[EMIM][Tf_2_N], [HMIM][Tf_2_N]	N_2_, CO_2_	[251]
12-hydroxystearic acid	[EMIM][Tf_2_N]/AgTf_2_N	**C_3_H_6_/C_3_H_8_**	[301]
**Network polymer/IL blends**			
Poly(dimethylacrylamide) (PDMAAm), poly(vinylpyrrolidone) (PVP), poly(2-hydroxyethylmethacrylate) (PHEMA), poly(methylacrylate) (PMA), poly(ethylacrylate) (PEA), +ethylene glycol dimethacrylate crosslinker	[P_4444_][Gly], [P_4444_][Ser], [P_4444_][Lys], [P_4444_][Pro]	N_2_, CO_2_ **CO_2_/N_2_**	[290]
(1) Poly(2-acrylamido-2-methyl-1-propanesulfonic acid) (PAMPS) (2) Polyvinylpyrrolidone (PVP) or polydimethylacrylamide (PDMAAm) + *N*,*N*’-methylenebisacrylamide (MBAA) crosslinker	[P_4444_][Pro], [P_2225_][Pro]	**CO_2_/N_2_**	[291]
(1) Poly(2-acrylamido-2-methylpropanesulfonic acid) (PAMPS) (2) Poly(dimethylacrylamide) (PDMAAm)	[P_4444_][Pro]	**CO_2_/N_2_ (H_2_O)**	[292]
(1) Poly(methacryloylamino propyl trimethylammonium chloride) (PMAPTAC) + *N*,*N*′-methylenebis(acrylamide) (MBAA) crosslinker (2) Poly(dimethylacrylamide) (PDMAAm)	[P_222(1O1)_][Inda]	**CO_2_/N_2_**	[293]
(1) 1,3-bis(2-oxiranyl-ethyl)imidazolium [Tf_2_N] (2) Tris(aminoethylamine) (TAEA)	[EMIM][Tf_2_N]	N_2_, CH_4_, CO_2_	[145]
	[EMIM][Tf_2_N], [EMIM][DCA]	**CO_2_/N_2_** (+H_2_O)	[188]
	[EMIM][Tf_2_N]	H_2_, He, N_2_, O_2_, CH_4_, and CO_2_ **CO_2_/CH_4_** (+H_2_O)	[189]
(1) Polydimethylacrylamide (PDMAAm) (2) *N*,*N*′-methylenebis(acrylamide) (MBAA) crosslinker (with silica nanoparticles, poly(TEOS))	[EMIM][Tf_2_N], [BMIM][Tf_2_N], [HMIM][Tf_2_N], [EMIM][B(CN)_4_], [BMIM][BF_4_], [P_2225_][Tf_2_N], [N_2225_][Tf_2_N]	**CO_2_/N_2_**	[295]
	[BMIM][Tf_2_N]	**CO_2_/N_2_** (+H_2_O)	[294]
Elvaloy4170, Reactive Ethylene Terpolymer	[EMIM][Tf_2_N]		[296]
(1) Ethoxylated (20) trimethylolpropane triacrylate(2) 3,6-dioxa-1,8-octanethiol	[EMIM][Tf_2_N], [HMIM][Tf_2_N], [iBMIM][Tf_2_N], [HEMIM][Tf_2_N], [C_3_C_N_MIM][Tf_2_N], [C_5_C_N_MIM][Tf_2_N], [TMSMIM][Tf_2_N], [BzMIM][Tf_2_N], [BzBIM][Tf_2_N], [EGMIM][Tf_2_N], [EG_2_MIM][Tf_2_N], [EG_3_MIM][Tf_2_N]	N_2_, CO_2_	[297]
(1) Poly(ethylene glycol) diacrylate (2) Poly(mercaptopropyl) methyl siloxane	[EMIM][AC], [EMIM][DCA], [EMIM][BF_4_], [EMIM][Tf_2_N], [BMPyrr][Tf_2_N], [EMPy][Tf_2_N], [P_1444_][Tf_2_N], [N_4111_][Tf_2_N]	N_2_, CO_2_	[299]
(1) Poly(ethylene glycol) diacrylate, Bisphenol A ethoxylate diacrylate and Trimethylolpropane ethoxylate triacrylate (2) Di(trimethylolpropane) tetraacrylate and dipentaerythritol penta-/hexa-acrylate	[EMIM][Tf_2_N]	N_2_, CO_2_	[298]
**Facilitated transport membranes**			
PIM-1	[BMIM]_2_[Co(NCS)_4_]	O_2_, N_2_	[268]
Poly([C_4_vim][Tf_2_N])	[C_4_MIM][Cl], [C_4_MIM][CuCl_2_]	CO_2_, H_2_, N_2_, CO **N_2_/CO**	[300]
Poly[pyridinium-1,4-diyliminocarbonyl-1,4-phenylene-methylene Tf_2_N^−^]	[EMIM][Tf_2_N]/AgTf_2_N	**C_3_H_6_/C_3_H_8_**	[301]
Poly[pyridinium-1,4-diyliminocarbonyl-1,4-phenylene-methylene PF_6_^−^]	[EMIM][BF_4_]/AgBF_4_	**C_3_H_6_/C_3_H_8_**	[301]
12-hydroxystearic acid	[EMIM][Tf_2_N]/AgTf_2_N	**C_3_H_6_/C_3_H_8_**	[301]

^(a)^ Single gases separated by a comma; mixed gases separated by “/” and printed boldface.

## 5. PIL/IL Blends

A special class of ion gel membranes is formed by PIL/IL blend membranes, for which the gel-forming polymer is a polymeric ionic liquid, whose ionic character favors higher stability of the gel in comparison with those prepared with conventional polymers. While the original scope of the development of poly(RTIL) gas separation membranes was to overcome several limitations of supported ionic liquid membranes (SILMs) and increase the selectivity, it was clear that a disadvantage is that they are up to 100 times less permeable than SILMs [159]. Therefore, early work on PILs has focused on increasing their permeability by polymerizing the IL monomers in the presence of non-polymerizable monomers [304,305,306], to incorporate the latter as plasticizers for the rigid neat PILs. Indeed, only 20 mol% of the free IL already led to a two- to five-fold increase in permeability, with no phase separation of the IL and the PIL [304,305], and operation was found to be stable in time [306]. A further increase in IL content up to 75 wt% was possible in cross-linked poly(vinylimidazolium) gel membranes, which reached impressive CO_2_ permeabilities of ca. 500 Barrer [307]. Meanwhile, many different systems have been investigated, including cross-linked PILs/ILs, facilitated transport PILs/ILs, and PIL/IL mixed matrix membranes [308], where the latter exceeded the upper bound for CO_2_/CH_4_ [309] in the case of mixed matrix membranes based on PIL/RTIL with SAPO-34 or SSZ-13 zeolites. The facilitated transport systems gave a particularly good performance for CO_2_/N_2_ separation with humidified gas streams, far beyond the 2008 upper bound, and TFC membranes yielded a CO_2_ permeance of 6100 ± 400 GPU [310]. The positive influence of water is likely due to a combination of physisorption and chemisorption through carbonation (formation of HCO_3_^−^/CO_3_^2−^) [311].

The majority of research on PIL/IL blend membranes, with extensive testing of different PIL and IL combinations, is focused on CO_2_ removal from nitrogen (e.g., in Post-combustion CO_2_ separation) [312], for which only PILs with sufficiently high molar mass yield suitable, mechanically robust self-standing films [313]. There is an optimum performance at intermediate molar mass because the CO_2_ permeability decreases if the molar mass becomes too high. Further studies involve the CO_2_ separation from other light gases such as CH_4_ and hydrogen [314]. Especially the latter seems to be a promising application field, in view of the clean production of biohydrogen as a renewable energy source, where the high solubility-selectivity of the ionic liquids favors the permeation of CO_2_ over H_2_ [142,315] and guarantees effective CO_2_ removal from H_2_ in multicomponent mixtures [316].

The most common PIL/RTIL composite membranes are based on a poly(vinylimidazolium) [307], pyrrolidinium polycation [312], poly(vinylbenzyl phosphonium) [314] backbone, but Gu and Lodge prepared a polystyrene-PIL-polystyrene block copolymer with a poly(2-bromoethyl acrylate) mid-block that was quaternized post-polymerization with 1-butylimidazole [150]. Imidazolium-mediated poly(benzoxazole) ionenes with alternating ionic and neutral units in the polymer backbone were reported by Bara’s group, who observed strong enhancement of CO_2_ permeability upon addition of ‘free’ ionic liquid [317].

## 6. Miscellaneous Uses of Ionic Liquids and Poly(ionic Liquids) Related to Gas Separation

Polymerization of ionic liquids vinyl, acrylate, or other polymerizable groups, results in the formation of ionic polymers, which are called poly (ionic liquid)s or polymerized ionic liquids (PILs) [111]. PILs can offer an advantageous combination of the electroconductive and thermostable properties with the mechanical stability of solid materials. Such a unique combination promoted their intensive research in electrochemistry (polyelectrolytes, batteries), surface chemistry (polymeric surfactants), or as hybrid nanocomposites, gas-sensors, and CO_2_-sorbents [107,241,242].

### 6.1. Mixed Matrix Membranes Incorporating ILs

A different use of ionic liquid in gas separation membranes, where they do not have a direct role in the gas transport, is that of a compatibilizer in mixed matrix membranes. Various groups studied the effect of RTILs on the separation performance of MMS based on zeolites such as SAPO-34 in polyethersulfone [318,319] or poly(RTIL) [163,309,320], on SSZ-13 in poly(RTIL) [309], on ZSM-5 in polyimide [321] or on Zeolite-4A in polysulfones [322]. In the latter case, 3- (trimethoxysilyl)propan-1-aminium acetate ([APTMS][Ac]) was covalently linked to the zeolite, increasing the CO_2_/CH_4_ and CO_2_/N_2_ selectivity around 45%, with a marginal effect on the CO_2_ permeability. In polymerized RTILs, the SAPO-34 zeolite and the RTIL synergistically improve both the CO_2_ permeability and the CO_2_/CH_4_ and CO_2_/N_2_ selectivity. This effect is ascribed to plasticization of the poly(RTIL) by the RTIL and on better contact between the SAPO-34 crystals and the more flexible polymer matrix [163]. In the SAPO-34/PES MMMs, the main effect is surface wetting of the filler by [EMIM][Tf_2_N], which improves the adhesion with the PES matrix [318,319].

For MOF-based mixed matrix membranes, the effect of the ionic liquids depends very much on the nature of the polymer matrix and can vary from interfacial toughening in ZIF-8/Pebax MMMs [323], to a homogeneous or quite heterogeneous distribution of the IL under certain conditions in ZIF-8/poly(RTILs) MMMs [324]. A comparative study of MIL-53(Al), Cu_3_(BTC)_2_, and ZIF-8 in the poly(ionic liquid) poly[Pyr_11_][Tf_2_N] with [C_4_mpyr][Tf_2_N] gives the highest CO_2_ permeability of 97 Barrer with 30 wt% of the ZIF-8 filler, slightly higher than the permeability reached with MIL-53(Al). Instead, the MIL-53(Al) yields systematically the highest CO_2_/H_2_ selectivity at all concentrations and reaches a maximum of 13.3 at a loading of 30 wt%. In the system ZIF-67/6FDA-durene, the ionic liquid reduces the nonselective interfacial defects and it also reduces the light gas adsorption in the filler, thus increasing the CO_2_/N_2_ and CO_2_/CH_4_ selectivity, and in particular the propylene/propane selectivity [325].

A strong synergistic effect of the IL and graphene oxide filler in Pebax^®^ 1657-based MMMs results in a 90% increase in CO_2_/N_2_ selectivity with a simultaneous increase of 50% in the CO_2_ permeability when the GO is grafted with ionic liquid at its surface [326]. In chitosan membranes with microporous titanosilicate ETS-10, the 1-ethyl-3-methylimidazolium acetate ionic liquid had the duplicate function to impart flexibility to the polymer matrix, and to heal defects around the nanoparticles to enhance compatibility [327]. Both additives synergistically improved the CO_2_ permeability and CO_2_/N_2_ ideal selectivity.

In nearly all these cases, the most important role of the ionic liquid is to reduce the nonselective defects at the polymer/filler interface, besides other effects, such as plasticization of the polymer matrix.

### 6.2. Porous Ionic Liquid Materials

The proper selection of the different polymerization methods of ILs can be used for tailoring the fully organic porous structure with porosities ranging from microporous to macroporous sizes [328]. Although the porosity can be achieved with a single linear polymer, these structures can immediately collapse. Thus, it is necessary to sufficiently stabilize the structure by a cross-linking agent. The overall rigidity of the network influences the final porosity with flexible parts, such as methylene bridges. Such a feature enables the rotation of the structure and can cause a partial collapse of the porosity. In that case, the material structure contains meso or macropores, and the total BET surface area is less than 200 m^2^ g^−1^. On the other hand, microporous materials with surface areas up to 1000 m^2^ g^−1^ can be prepared by templating or templating-free methodologies. In the templating method, the material template is filled with a polymer or monomers to be polymerized, and, subsequently, the template is chemically removed. The resulting structure exhibited meso- or macro-size porosity. While the templating-free methodologies are based on polymerization with multitopic and sufficiently stiff monomers, either by free radical polymerization of ILs monomers or by a direct synthesis method when the ionic liquid moieties can be generated during the formation of the polymer network. When the packing of the polymer network is inefficient due to the stiff polymer network, sufficiently stable micropores can be formed. An alternative simple method for meso/macroporous polymer network preparation is polyelectrolyte complexation of PILs chains.

### 6.3. Gas Detection

Changes in electroconductivity of PILs upon absorption of certain gases or vapors can be utilized for a specific gas or vapor detection. The majority of the studied cationic PILs employed [Tf_2_N], [BF_4_], [PF_6_], and [Cl] anions for sensing applications such as detecting food aromas, such as beer volatiles, coffee aroma, nitroaromatic explosives, or humidity sensors [241]. Although the membrane does not act as a selective barrier for mass transport in this application, the operation principle depends on the same features, especially on the gas solubility in the polymer.

### 6.4. CO_2_ Absorption and Capturing

The effects of structural variation in PILs, the choice of cation, anion, backbone, alkyl chain length, porosity, and cross-linking on the CO_2_ absorption and capturing were thoroughly reviewed in several works [107,241,242,243]. In summary, the cations play a decisive role in CO_2_ solubility in cationic PILs obtained via radical polymerization. Studies reporting the use of PILs synthesized via condensation polymerization and polymer modification for CO_2_ capture and separation are still less frequent compared to those synthesized by radical polymerization [242]. Further, the type of cation plays a key role in defining the PILs features, in contrast to RTILs, in which anions are more imperative. Improved performance among PILs was reported for those

(i)with ammonium and imidazolium cations and [Tf_2_N], [BF_4_], [PF_6_], [DCA], and acetate anions,(ii)rigid polystyrene backbone,(iii)high surface area and porosity, together with high separation efficiency [243].

For PILs prepared by radical polymerization, the observed CO_2_ sorption is higher in the tetraalkylammonium cations PILs when compared to the imidazolium cation, due to higher density of positive charges in the tetraalkylammonium cation, resulting in strong interaction with CO_2_, despite the positive imidazolium charge is delocalized [244]. Furthermore, in cationic PILs, the cross-linking and the longer alkyl side chains in the cations hinder CO_2_ sorption due to steric effects and lead to decreasing the sorption capacity. For styrene-based PILs with [BF_4_] as the counter anion, CO_2_ sorption capacity decreases in the following order: Ammonium > pyridinium > phosphonium > imidazolium [247].

Although the anion has less influence than the cation on the use of PIL as a sorbent for CO_2_ capture, it still needs to be considered. Various inorganic (Br-, BF_4_-, PF_6_-, Tf_2_N-, etc.) and organic (Sac, Bz, Ac, etc.) anions have been reported in the literature for obtaining cationic PILs [242]. Different trends were observed for PILs compared to room temperature ILs, for instance, fluorinated anions in PILs show no direct impact on CO_2_ sorption, and among all PILs evaluated, the CO_2_ solubility for PILs increased in the following order, [TFAc] < [HFB] < [Bz] < [Ac], according to the increase in basicity (pKa of the conjugate acid) [329]. Furthermore, PILs obtained via condensation polymerization are mainly based on poly(ethylene oxide), polyurethane polybenzimidazoles, polyimides, and imidazolium containing ester and hydroxyl groups [242].

Generally, the principal idea to use PILs as materials for CO_2_ capture was based on knowledge of RTILs properties, e.g., high CO_2_ uptakes compared to other gases, and its reversible sorption/desorption [107]. Pioneer works on PILs proved that such a concept is feasible because PILs have a high CO_2_ absorption capacity [244]. Calculated Henry’s constants via extrapolating the slope to the zero CO_2_ partial pressure of polymerized 1-(p-vinylbenzyl)-3-butyl-imidazolium tetrafluoroborate (p[VBBI][BF_4_]) and (1-[2-(methacryloyloxy)ethyl]-3-butyl-imidazolium tetrafluoroborate (p[MABI][BF_4_]) were 26.0 bar and 37.7 bar at 22 °C. These are, however, lower than that of room temperature ionic liquid [BMIM][PF6] (53.4 bar at 25 °C). Notwithstanding, in 2009, Radosz and Shen patented PILs with imidazolium-, phosphonium-, ammonium-, and pyridinium-based polycations as efficient selective CO_2_ sorbent materials [246]. The structural variations of the cation, e.g., increasing the length of the *N*-alkyl substituent or introducing polar substituents, were found useful for improving CO_2_ permeability and selectivity. The PILs series based on poly[1-(para-vinylbenzyl)-triethylammonium cation such as p[VBTMA][BF_4_], p[VBTMA][PF_6_], and p[VBTMA][Tf_2_N] and poly[2-(methacryloyloxy)ethyl]trimethylammonium tetrafluoroborate (p[MATMA][BF_4_]) revealed higher CO_2_ sorption of styrene backbone PILs compared to polymethylmethacrylate backbone due to higher styrene rigidity [248]. Further, p[VBTMA][PF_6_] exhibited an outstanding 77% (*w*/*w*) CO_2_ adsorption capacity and increasing gas sorption values while the temperature decreased, thus proving the physisorption nature of the adsorption process [249]. The effect of substituent size on the sorption properties of polymerized 1-vinyl-3-ethylimidazolium dicyanamide (p[VEIM][DCA]), 1-vinyl-3-butylimidazolium dicyanamide (p[VBIM][DCA]), and 1-vinyl-3-heptylimidazolium dicyanamide (p[VHIM][DCA]) was reported by Li et al. [222]. PILs with a longer *N*-alkyl group have a higher free volume than those with shorter *N*-alkyl groups, thus the gas solubility (CO_2_ and N_2_) increases in the order of (p[vhim][DCA]) > poly([VBIM][DCA]) > poly([VEIM][DCA]). Determined CO_2_–N_2_ solubility-selectivities were 8.6–12.9, less than the solubility selectivity of [EMIM][DCA]. The length of the *N*-alkyl group was found to be responsible for the dilution of the concentration of polar groups (ions). Therefore, PILs favor to also absorb non-polar nitrogen, which resulted in a decrease of the overall CO_2_–N_2_ solubility-selectivities. Thereafter, Kumbharkar et al. prepared PILs based on three *N*-quaternized polybenzimidazoles [TMPBI] and 5-*tert*-butyl isophthalic acid (BuI) with different anion counter ions [231]. The resulting gas sorption isotherms were 2–3 times higher than those of polysulfone (PSF) and polycarbonate (PC). The reported CO_2_ solubility coefficients (2.18–4.59 cm^3^_(STP)_ cm^−3^ atm^−1^) and sorption selectivities for CO_2_/N_2_ (7.13–19.60) and CO_2_/CH_4_ (4.89–13.08) revealed that increasing of CO_2_ sorption uptakes and selectivities in the order [Tf_2_N] < [BF_4_] < [Ac] correspond with increasing of anion basicity. The same group also studied pyrenyl and anthryl functionalized PILs with *tert*-butylpolybenzimidazole [PBI-BuI]. Synthesized PILs showed increasing CO_2_ sorption in the order of anion variation as [Tf_2_N] < [Br] < [BF_4_], whereas [BF_4_] also exhibited the highest dual-mode sorption parameters. The favored CO_2_ interactions for the [BF_4_] anion were found to correlate with the basicity of the anions, while the role of the bulk of substituent remained subordinate. The effect of the anion was found less significant for sorption of other gases (H_2_, N_2_ and CH_4_) compared to CO_2_. Most probably they were governed by usual properties associated with the glassy nature of these PILs [232]. Further, Einloft et al. [330] compared the effect of the backbone (aliphatic vs. aromatic) with the same anion and the counter cation. Usually, the presence of aromatic groups in the polymer backbone significantly improves the CO_2_ uptake, but only at low gas pressures. Recently, a condensation of various diisocyanates with novel ionic diamines and subsequent ion metathesis reaction was studied by Morozova et al. [141]. The resulting PILs exhibited remarkably high CO_2_ capture (10.5–24.8 mg g^−1^ at 0 C and 1 bar). The CO_2_ sorption was found to be dependent both on the nature of the cation and on the structure of the diisocyanate. The highest sorption was observed for tetrafluoroborate polyurethane-based PIL with 4,4-methylene-bis(cyclohexyl isocyanate) diisocyanate and aromatic diamine bearing quinuclidinium (24.8 mg g^−1^ at 0 C and 1 bar) and dicycloaliphatic (24.8 mg g^−1^) cations.

Summarizing, over the last decade, PILs have emerged as an effective and sustainable CO_2_ capture platform. Tailored structure and suitable properties add an important value to novel design strategies of efficient CO_2_ sorbents. Continuous improvement of the sorption properties of PILs caused the transformation from scientific curiosity to future technological interest in creating new efficient CO_2_ sorbents.

## 7. Conclusions

Ionic liquids in their various forms represent a new family of materials that provide an excellent starting point for the development of innovative membranes for gas and vapor separation. The enormous variety of available anions and cations allows tailoring of the IL properties to the needs of the envisaged separation process. The introduction of special functional groups yields task-specific ionic liquids or carriers for facilitated transport membranes. In their traditional form, as supported ionic liquid membranes impregnated in suitable porous membrane substrates, they provide a relatively high permeability, but moderate pressure-resistance. Various experimental factors affect their performance, such as the ionic nature, the viscosity, the presence of humidity, as well as the operating temperature and pressure. Careful control of these factors allows the preparation of membranes with a performance that exceeds the Robeson upper bounds for various gas pairs, such as CO_2_/N_2_. SILMs have a relatively poor pressure resistance and are therefore most suitable for low-pressure processes. Key achievements in using SILMs in gas separation compared to other techniques or conventional membranes are low solvent use, higher permeability and often also higher selectivity than polymer membranes, high interfacial area per unit volume for mass transfer, and more efficiency in application over other liquid membrane processes.

Polymer/IL blends or ion-gel membranes offer a better pressure resistance, as they were designed to immobilize the ILs in a much tighter network of the gel-forming polymer chains. These gels are simplest to prepare from conventional semi-crystalline polymers by a solvent evaporation method, using a common solvent for both the polymer and the ionic liquid. This leaves a physically cross-linked gel, where the crystal domains serve as cross-links. Analogously, segmented block-copolymers can be used, which self-assemble into micro-phase-separated gels with amorphous and/or semi-crystalline microdomains. Both systems of semi-crystalline polymers and segmented block-copolymers yield reversible physical networks. Instead, chemically cross-linked polymers are typically formed by polymerization of mono-functional vinylic or acrylic ionic liquid monomers in the presence of divinyl- or diacrylate cross-linker, or by the curing of epoxy-functional ionic liquids with multi-functional amines. These systems may provide lower leaching of the ionic liquid, while excess free amine groups in the cross-linked ion-gels provide fixed-site carriers for facilitated transport of for instance CO_2_. Although they are more pressure-resistant than the SILMs, the ionic liquid may still be subject to unwanted leaching from the blend.

The need to suppress the leaching of ionic liquids from the membrane stimulated the development of polymerized ionic liquids in which the ionic moiety is covalently linked to the polymer backbone. A broad spectrum of different polymers and incorporated ionic moieties provides polymers with a wide range of properties. Unfortunately, the resulting polyelectrolytes form rather rigid dense membranes with low permeability. A highly effective way to increase their permeability is by blending with free ionic liquid, and a successful way to achieve this is by polymerization of polymerizable ionic liquids in the presence of non-polymerizable ionic liquids. The PIL/IL combination offers superior stability compared to the blends from conventional neutral polymers when properly chosen, and with excellent transport properties that can exceed the 2008 Robeson upper bound for important gas pairs, such as CO_2_/N_2_. The most important parameters to influence the gas transport properties of the conventional polymer/IL blends and the PIL/IL blends are the type of polymer and IL, and in particular the final concentration of IL in the blend.

Other uses where the IL successfully improves the membrane performance are in mixed matrix membranes, where ILs may increase the permeability, the selectivity, or both, synergistically with the porous organic, inorganic, or metalorganic filler. The ionic liquid can be used mainly to reduce the nonselective interface defects between the zeolite or MOF filler particles and the polymer matrix, and to improve the bulk properties of the polymer matrix, for instance by plasticization.

### Further Outlook

Summarizing the numerous studies reported so far, it can be concluded that ionic liquids in their various forms offer an excellent platform for the development of effective gas separation membranes with a wide range of chemical, physical, and transport properties. While the performance of dense films has been extensively explored, predominantly with singles gases, providing permeability and ideal selectivity data, there are still opportunities for more studies of the mixed gas separation performance under realistic operation conditions at different temperatures, pressures, and gas compositions. In particular, the effect of trace amounts of impurities, such as humidity, H_2_S, hydrocarbon vapors, etc., needs to be studied further. The preparation and performance of thin-film composite membranes is also a topic that needs further exploration, both as flat films and as composite hollow fiber membranes. Finally, the use of ionic liquids to act as a carrier for facilitated transport offers important challenges for more and more efficient and selective separation processes for the capture of low concentrations of CO_2_ from the air, or difficult olefin/paraffin separations.

## Figures and Tables

**Figure 1 membranes-11-00097-f001:**
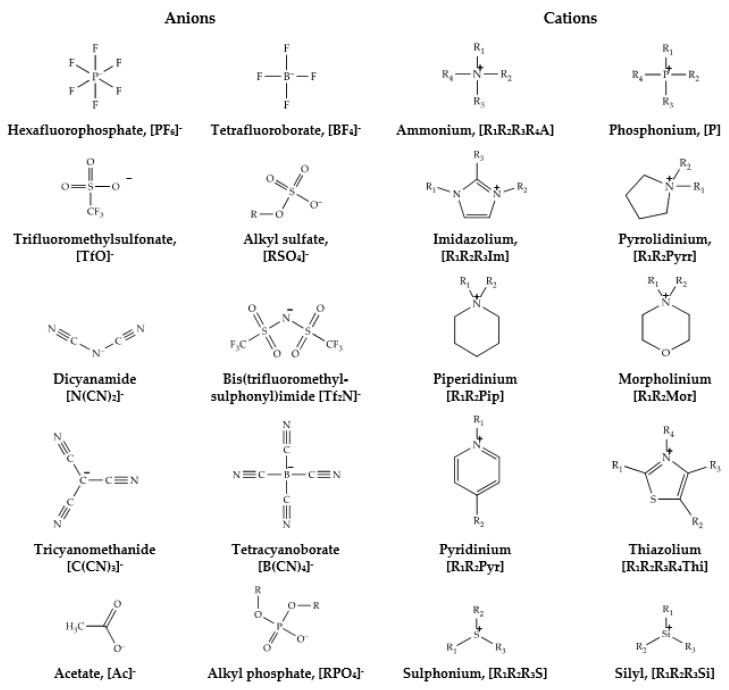
Overview of some of the most common anions and cations used in ion ionic liquid membranes.

**Figure 2 membranes-11-00097-f002:**
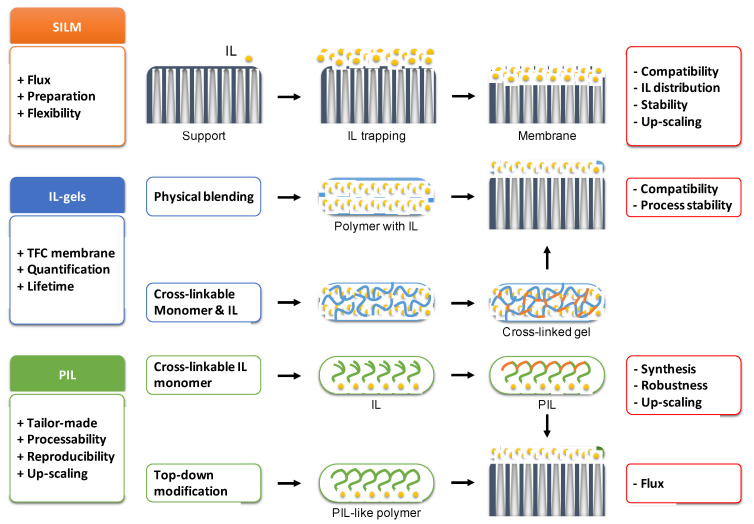
Schematic summary of the membrane characteristics and preparation techniques employing ionic liquid (IL)-based materials in membrane-based gas separation.

**Figure 3 membranes-11-00097-f003:**
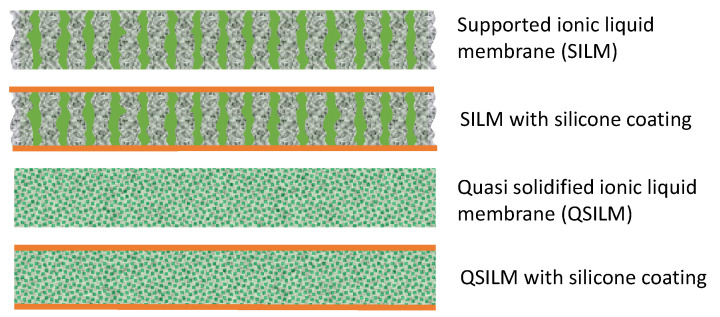
Schematic representation of the possible types of supported ionic liquid membranes. Green = ionic liquid, grey = porous support of gel former, orange = silicone.

**Figure 4 membranes-11-00097-f004:**
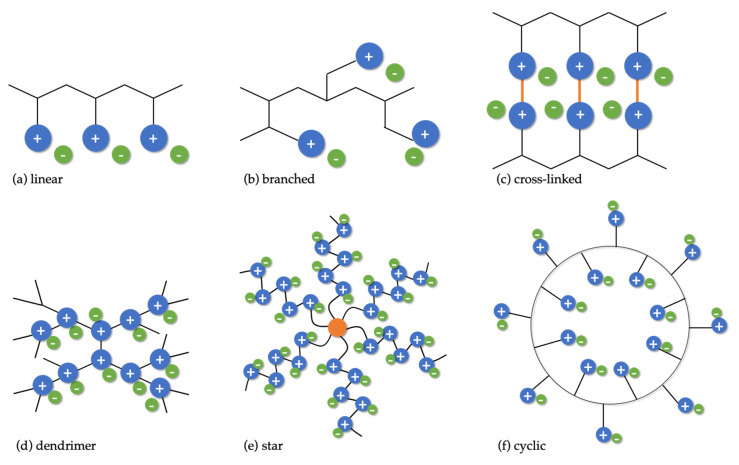
Schematic overview of the different architectures of polymeric ionic liquids (PILs), based on polycations. Blue spheres: cationic moieties; green spheres, anions. Analogous structures are possible with polyanions.

**Figure 5 membranes-11-00097-f005:**
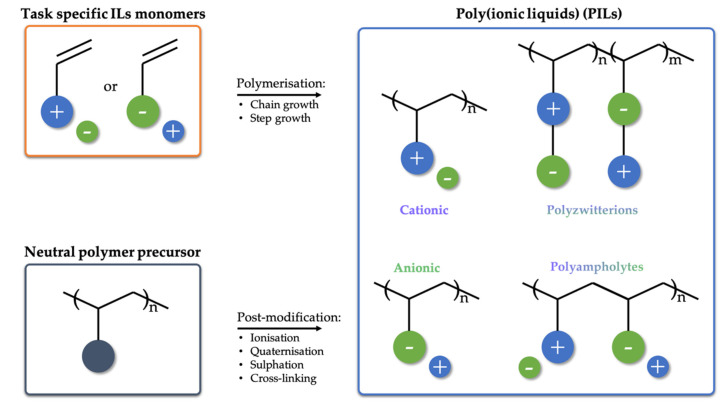
Poly(ionic liquids) synthesis via polymerization of task-specific IL monomers (**top-left**) and post-modification of polymeric precursors (**bottom-left**) yields similar polymeric structures based on their ionic composition: Cationic (**top-center**), anionic (**bottom-center**), polyzwitterions (**top-right**), and polyampholytes (**bottom-right**). Ions (small spheres) or ionic moieties (large spheres) are indicated with the positive (blue +, cation) or negative (green -, anion) symbol.

**Figure 6 membranes-11-00097-f006:**
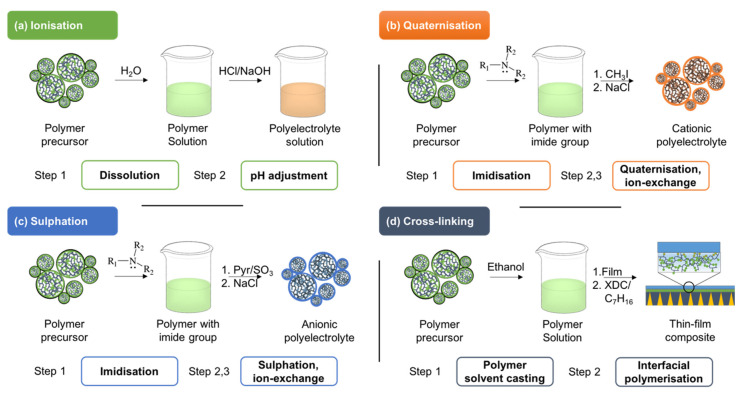
Major approaches for post-synthetic modification of polymers to produce ‘top-down’ polyelectrolytes include (**a**) ionization, (**b**) quaternization, (**c**) sulphation, and (**d**) cross-linking. XDC abbreviates p-Xylylene dichloride.

**Figure 7 membranes-11-00097-f007:**
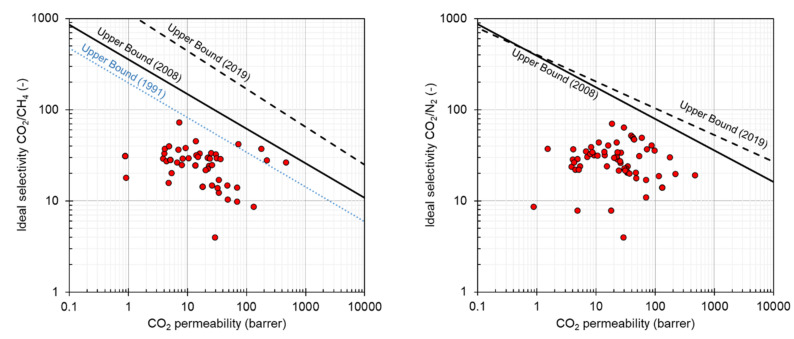
Robeson plots for CO_2_/CH_4_ (left) and CO_2_/N_2_ (right) for PILs (Table 5) with corresponding upper bounds from 1991 [238], 2008 [239], and 2019 [240]. (1 Barrer = 10^−10^ cm^3^_STP_ cm cm^−2^ s^-1^ cmHg^−1^ = 3.348 × 10^−16^ mol m m^−2^ s^−1^ Pa^−1^).

**Figure 8 membranes-11-00097-f008:**
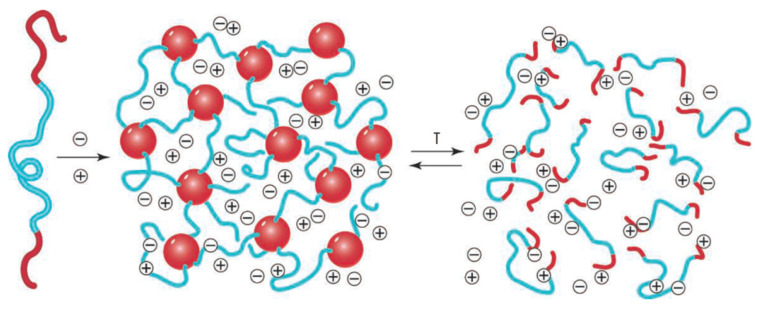
Thermally reversible ion gel based on an ABA triblock copolymer. The blue block is soluble in the ionic liquid, indicated as (+)(−), and the red block is insoluble at room temperature and becomes soluble upon heating. Reprinted from Ref. [252] with permission from AAAS.

**Figure 9 membranes-11-00097-f009:**
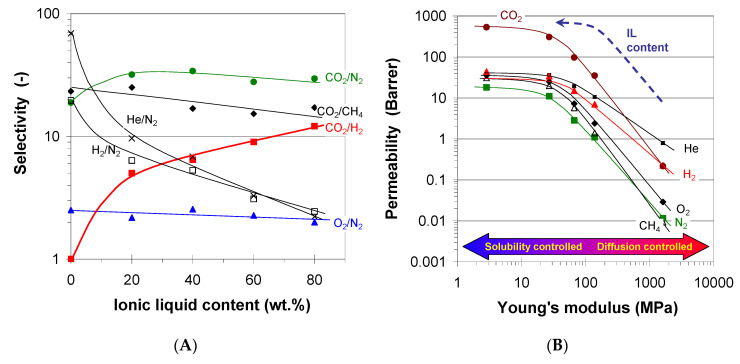
(**A**) Correlation of the selectivity and the ionic liquid content and (**B**) correlation of the permeability and Young’s modulus for p(VDF-HFP)/[EMIM][Tf_2_N] membranes with IL content ranging from 0% to 80%. 1 Barrer = 10^−10^ cm^3^_STP_ cm cm^−2^ s^−1^ cmHg^−1^ = 3.348 × 10^−16^ mol m m^−2^ s^−1^ Pa^−1^. Reprinted from [256]. Copyright (2012), with permission from Elsevier.

**Figure 10 membranes-11-00097-f010:**
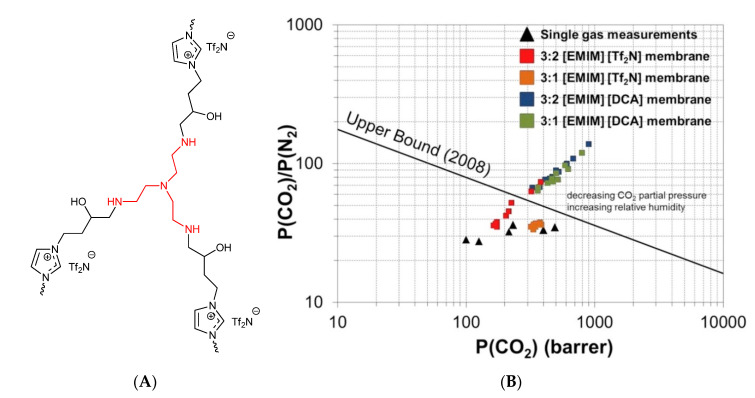
(**A**) Network structure for stoichiometry 3:1, assuming 100% conversion, and (**B**) CO_2_/N_2_ Robeson plot (right) with data for epoxy-amine ion gels tested with single gases (▲) [145] and with humidified mixed-gas feeds (□) [188], where the ratios 3:1 and 3:2 refer to the mole ratio of the epoxy monomer and the amine monomer. 1 Barrer = 10^−10^ cm^3^_STP_ cm cm^−2^ s^−1^ cmHg^−1^ = 3.348 × 10^−16^ mol m m^−2^ s^−1^ Pa^−1^. Robeson plot reprinted from [188]. Copyright (2015), with permission from Elsevier.

## Data Availability

Data are available from the original manuscript in the references.

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
