# Peer review of "A Review on Ionic Liquid Gas Separation Membranes"

_membranes, 2021, doi:10.3390/membranes11020097_

Round 1

Reviewer 1 Report

This review comprehansively reviewed the recently ionic-liquid based membranes for gas separation. The paper is well organized and could be accepted after minor revision.

  1. The unit for t in Table 5 is wrong.
  2. It is better to give a summary for the separation performance of the reported membrane with a upper bound line both in Barrer and GPU. This will be very helpful for the new comers to recoganize this field clearly.
  3. Please give some comments for the performance in Barrer or GPU?

Author Response

Comments and Suggestions for Authors

This review comprehansively reviewed the recently ionic-liquid based membranes for gas separation. The paper is well organized and could be accepted after minor revision.

  1. The unit for t in Table 5 is wrong.

T was capitalized (temperature) and the degrees symbol was added (°C)

  1. It is better to give a summary for the separation performance of the reported membrane with a upper bound line both in Barrer and GPU. This will be very helpful for the new comers to recoganize this field clearly.

We thank the reviewer for this suggestion. As an example, we have added an additional figure (new Figure 7 in the revised manuscript) with two Robeson plots for CO2/N2 and CO2/CH4 of a number of PILs from Table 5. Another example of a Robeson plot for a specific system was already given in Figure 9 (Figure 10 in the revised manuscript).

Unfortunately, it is hardly feasible to plot the analogous data of all membranes described in this manuscript, because they are too numerous, and in many cases, a single membrane is tested under many different conditions of pressure, temperature, humidity, gas mixture composition, leading to an unfair comparison.

For similar reasons, we did not include an alternative graph of the selectivity as a function of the permeance. Of course, the direct comparison of the separation performance of membrane materials in Barrer and in GPU units directly in one graph is impossible, because their conversion requires division/multiplication by the sample thickness (1 GPU = 1 Barrer / 1 micrometre of the membrane thickness (i.e. thickness of the separation layer). For thin-film composite membranes, asymmetric membranes or composite SILMs, the latter is often not available.

  1. Please give some comments for the performance in Barrer or GPU?

Generally, both units are used to evaluate the membrane performance but they are not listed in the SI system of units. The Barrer unit is used as the permeability unit while the GPU (gas permeance unit) is used to express the permeance. Further, permeance characterizes gas transport through the membrane as pressure-normalized steady-state flux and permeability is calculated by multiplying permeance by the thickness of the (homogeneous) membrane (and this applies 1 Barrer / 1 micron = 1 GPU). Usually, permeance is used for those membranes (e.g. composite or multilayer) with the unknown thickness of their separation layer.

[Drioli E., Giorno L. (eds) Encyclopaedia of Membranes. Springer, Berlin, Heidelberg. https://doi.org/10.1007/978-3-642-40872-4_260-1 ]. Both units were introduced for easier understanding of the results because the values of both quantities in common SI units are usually very small numbers.

To explain the concept, the following text is added to section 2.2 where we first speak about transport properties:

“It must be noted that the transport properties of SILMs are often expressed as permeance (unit GPU). Whereas, for homogeneous membranes the permeability is usually defined as the permeability coefficient, a material-specific property, normalized for the thickness (with unit Barrer, where 1 Barrer = 10‑10 cm3STP cm cm‑2 s‑1 cmHg‑1 = 3.348 ∙ 10-16 mol m m-2 s-1 Pa‑1), this is not possible for non-homogeneous or composite materials with undefined or unknown thickness, such as SILMs or thin-film composites. The permeance is usually expressed in GPU ‘Gas Permeance Unit’, where 1 GPU = 1 Barrer µm‑1 = 10‑6 cm3STP cm‑2 s‑1 cmHg‑1 = 3.348 ∙ 10-10 mol m-2 s-1 Pa‑1. Alternatively, it is often expressed in m3STP m‑2 h‑1 bar‑1. Permeance is a membrane property and not a material property”

The unit of Barrer and GPU were also added to the footer of the tables or to the figure caption where they occur.

1 Barrer = 10‑10 cm3STP cm cm‑2 s‑1 cmHg‑1 = 3.348 ∙ 10-16 mol m m-2 s-1 Pa‑1

1 GPU = 1 Barrer µm‑1 = 10‑6 cm3STP cm‑2 s‑1 cmHg‑1 = 3.348 ∙ 10-10 mol m-2 s-1 Pa‑1

Reviewer 2 Report

I was glad to review this manuscript. The work summarized and critically evaluates the aspects of the aimed topic. In my opinion, this work could be a nice contribution to the scientific outputs in the field of SILMs and gas separation. My minor comments are the followings:

P4 L122 'SLIMs'

P4 L 131-132: rephrase the sentence.

P4 L138: rephrase the sentence.

I suggest to include a list of abbreviations, and - considering the length of the manuscript - a table of content.

Thoroughly double-check the text for typos and proper grammar.

Author Response

Reviewer 2

Comments and Suggestions for Authors

I was glad to review this manuscript. The work summarized and critically evaluates the aspects of the aimed topic. In my opinion, this work could be a nice contribution to the scientific outputs in the field of SILMs and gas separation. My minor comments are the followings:

P4 L122 'SLIMs'

The typo was corrected

P4 L 131-132: rephrase the sentence.

The text was rephrased as: “For example, SILMs prepared by the vacuum method show higher losses of the IL during operation, and the relatively losses increase with increasing viscosity of the IL.”

P4 L138: rephrase the sentence.

The text was rephrased as: “The IL easily diffused into the small pores, helped by the capillary condensation in the presence of ethanol besides supercritical CO2, leading to the selective filling of the meso- and micropores in the supports”

I suggest to include a list of abbreviations, and - considering the length of the manuscript - a table of content.

As suggested by the reviewer, we added a table of contents and a list of Symbols and abbreviations to the manuscript.

Thoroughly double-check the text for typos and proper grammar.

We thoroughly checked the entire manuscript. Minor changes were made throughout the text but were not highlighted in the revision to avoid too much confusion.

Reviewer 3 Report

Specific Comment.

Page 3, lines 67, 68 and lines 76-78, presents the same idea or proposal, so in order to avoid repetition it is suggested to rewrite it and present the idea in a single sentence

Author Response

Reviewer 3

Comments and Suggestions for Authors

Specific Comment.

Page 3, lines 67, 68 and lines 76-78, presents the same idea or proposal, so in order to avoid repetition it is suggested to rewrite it and present the idea in a single sentence

We are sorry that our intention was not clear, but line 67,68 briefly summarize the contents, whereas lines 76-78 indicate the objective of the manuscript.

The objective is slightly rewritten to make this point more clear : “The present manuscript aims to provide a comprehensive overview of the current state of the art in ionic liquid-based membranes for gas separation applications, to highlight the pros and cons of each of these three different categories of ILMs, and to compare their performance.”
